# Efficient Online Influence Maximization under the Independent Cascade Model with Node-Level Feedback

**Arpit Agarwal** [1]    **Varad Deolankar** [2]    **Rohan Ghuge** [3]

## Abstract

Influence maximization is an important research area in social network analysis, where the goal is to select a small set of seed nodes so as to maximize the expected spread of influence under a stochastic diffusion process. Classical approximation algorithms for this problem rely on full knowledge of the underlying influence probabilities and operate in an offline manner. In many real-world settings, however, these probabilities are unknown and must be learned from data, raising the question: *can one still obtain strong performance guarantees while simultaneously learning the diffusion model parameters through repeated interactions?* In this paper, we study the problem of *online influence maximization* under the independent cascade model, where influence probabilities are unknown and feedback is limited to *node-level* activation outcomes. Prior work relies on a *pair oracle* which needs to perform a joint optimization over seed sets and feasible parameters. This oracle is difficult to implement in practice and it was open whether one can achieve sublinear regret using only a *standard* offline oracle. We resolve this question by designing an online learning algorithm that achieves $\widetilde{O}(\sqrt{T})$ regret using only a *standard* offline oracle. Finally, we validate our theoretical results via experiments on real and synthetic data.

## 1. Introduction

Consider a social network in which individuals influence each other with certain probabilities. Starting from a small

set of initially activated "seed" nodes, influence spreads through the network in discrete steps: activated nodes may activate their neighbors, and those newly activated nodes may in turn activate others. The influence maximization problem asks how to select a seed set of size $k$ so as to maximize the expected total number of activated, or *influenced* nodes at the end of the diffusion process. This problem arises in many practical settings. For example, in viral marketing one may want to maximize product adoption by targeting a small number of early users (Huang et al., 2019); in opinion diffusion, a campaign may seek to maximize the spread of a message (Castiglioni et al., 2021); and in epidemic control, the goal may be to understand the spread of the infectious disease so as to design interventions that reduce the final outbreak size (Yao et al., 2022).

A central component of the influence maximization problem is the underlying diffusion process that governs how influence propagates through the network. A variety of stochastic diffusion models have been studied in the literature, including the independent cascade model, the linear threshold model, and their many variants and generalizations (Kempe et al., 2003; Chen et al., 2010). The seminal work of Kempe et al. (2003) showed that under both the independent cascade and linear threshold models, the expected influence spread is a monotone submodular function of the seed set. This implies that a canonical greedy algorithm achieves a tight $(1 - 1/e)$-approximation, and no polynomial-time algorithm can do better unless P = NP (Feige, 1998). As a result, much of the early literature (Leskovec et al., 2007; Chen et al., 2009; Goyal et al., 2011) focused on *offline* influence maximization in which the diffusion parameters are assumed to be known in advance and the algorithm selects a single, fixed seed set in a non-adaptive manner. More recently, there has been growing interest in understanding the role of *adaptivity* in influence maximization, where the algorithm is allowed to react to observed diffusion outcomes when selecting future seeds. A sequence of works has investigated adaptive seeding strategies and quantified the potential gains from adaptivity (Golovin & Krause, 2017; Peng & Chen, 2019; Chen et al., 2022).

All of the works discussed above assume that the underlying influence probabilities governing the diffusion process

[1]Department of Computer Science & Engineering, Indian Institute of Technology Bombay, Mumbai, India. [2]Department of Marketing, National University of Singapore, Singapore. [3]Department of Information, Risk, and Operations Management, University of Texas at Austin, Austin, USA. Correspondence to: Rohan Ghuge <rohan.ghuge@mccombs.utexas.edu>.

*Proceedings of the 43$^{rd}$ International Conference on Machine Learning*, Seoul, South Korea. PMLR 306, 2026. Copyright 2026 by the author(s).

are known a priori. In practice, however, these probabilities are rarely available and must instead be learned from data. This raises a natural question: *can we design efficient algorithms for influence maximization when the diffusion parameters are unknown but can be learned through repeated interactions with the network?* This setting gives rise to the problem of *online influence maximization*, in which an algorithm repeatedly selects seed sets over multiple rounds, observes feedback from the resulting diffusion processes, and aims to simultaneously learn the influence probabilities and perform well over time. A key consideration in this online setting is the *type of feedback* available to the learner. Under *edge-level feedback*, where the learning algorithm observes which specific edge caused the activation, several recent works obtain $\widetilde{O}(\sqrt{T})$ regret guarantees using upper-confidence-bound–based methods (see, for example, (Chen et al., 2016), (Wang & Chen, 2017) and (Wen et al., 2017)).

However, edge-level feedback is often unrealistic in practical applications. Moreover, empirical studies of network formation suggest that real-world social networks are often dense (Johnsson & Moon, 2021), so nodes may have many active neighbors; consequently, even when a node becomes active, it is typically impossible to infer which incoming edges were live during the diffusion process. For instance, while one may observe that a user adopts a product, it is typically unclear which of their in-neighbors persuaded them to adopt. Motivated by these considerations, a growing line of work has studied online influence maximization under *node-level feedback*, where the learner observes only which nodes become active (and at which times) but not the identities of the influencing edges (Vaswani et al., 2015; Li et al., 2020; Zhang et al., 2022). A related adversarial framework of *sum-max submodular bandits* (Pasteris et al., 2024) is also applicable to online influence maximization via a live-edge reduction; we discuss the relationship and incomparability of guarantees in Appendix A.

The absence of edge-level information introduces significant challenges in estimation, as the influence probabilities are only partially identifiable from node-level observations, and multiple parameter settings may be consistent with the observed data. Prior work relies on a powerful abstraction, i.e. *pair oracles*, which can jointly optimize over seed sets and feasible parameters (Li et al., 2020; Zhang et al., 2022).[1] However, such oracles are significantly more powerful than standard offline oracles and are not known to admit efficient implementations. In light of these considerations, the main motivating question of our work is the following:

*Is there an algorithm for online influence maxi-*

---

*mization under the independent cascade model with node-level feedback achieving $\widetilde{O}(\sqrt{T})$ regret using a standard offline oracle?*

In this work, we answer this question in the affirmative. We design an online learning algorithm for influence maximization under the independent cascade model that operates under node-level feedback and achieves $\widetilde{O}(\sqrt{T})$ regret using only a standard offline oracle.

Our main technical contribution is to overcome the challenge of partial identifiability in estimating edge-level probabilities by instead focusing on *set-level* aggregate parameters that remain identifiable from node-level feedback. Although the number of parameters per node grows exponentially with its in-degree, we show that efficient estimation is possible by leveraging a generalized linear structure. However, unlike prior work that relies on pair oracles (Zhang et al., 2022), we construct separate upper confidence bounds (UCBs) for each set-level activation probability and show that a standard offline oracle can compute optimistic seed sets given these values. Using the principle of optimism in the face of uncertainty, we bound regret in terms of the widths of these confidence intervals. Finally, to handle the stochasticity of the diffusion process, we establish a stability lemma that bounds regret in a pay-per-use fashion, attributing regret to parameters proportionally to their 'expected usage' during diffusion (see Section 1.2 for more details).

Not only do we achieve $\widetilde{O}(\sqrt{T})$ regret, but our algorithm also runs in polynomial time using a standard offline oracle such as the the greedy algorithm of Kempe et al. (2003). Moreover, compared to the pair-oracle-based approach of Zhang et al. (2022), our method significantly improves the dependence on key problem parameters, including $\gamma$, a lower bound on node inactivity probabilities, and $n$, the number of nodes in the network (see Section 1.2 for a detailed comparison). Finally, we complement our theoretical analysis with experiments on both real and synthetic networks, demonstrating that our learning algorithm performs competitively against existing baselines.

### 1.1. Problem Setup

We first set up notation for the influence maximization problem in the known-distribution setting, and then describe the online learning framework studied in this paper.

**Influence Maximization.** At a high level, an instance of *influence maximization* consists of an underlying graph, a stochastic diffusion model governing how influence propagates through the graph, and a budget $k \in \mathbb{N}$. The objective is to select a seed set of at most $k$ nodes so as to maximize the expected number of nodes influenced by the end of the diffusion process.

Formally, let $G = (V, E)$ be a directed graph with $|V| =$

$n$ nodes and $|E| = m$ edges. Each edge $e = (u, v) \in E$ is associated with an *influence probability* $p_e \in (0, 1)$, representing the probability that node $u$, once influenced, successfully influences node $v$. We denote the vector of influence probabilities by $\mathbf{p} = (p_e)_{e \in E}$. We will also use a shorthand $p_{uv}$ to denote the influence probability of edge $e = (u, v)$ when the endpoints need to be made explicit. For each node $v \in V$, let $N(v) = N^{\text{in}}(v)$ denote the set of in-neighbors of $v$, and let $d_v = |N(v)|$ be its in-degree. We write $D = \max_{v \in V} d_v$ for the maximum in-degree of $G$.

As mentioned before, influence maximization has been studied under many diffusion models. In this paper, we focus on the *independent cascade* (IC) model. Under the IC model, diffusion proceeds in discrete time steps $\tau = 0, 1, \ldots, n-1$. Let $S_\tau \subseteq V$ denote the set of nodes newly activated at time $\tau$. Initially, at time $\tau = 0$ a seed set $S_0 \subseteq V$ is activated. At each subsequent step $\tau \geq 1$, every node $u \in S_{\tau-1}$ independently attempts to activate each inactive neighbor $v \in V \setminus \bigcup_{\tau'=0}^{\tau-1} S_{\tau'}$ with probability $p_{uv}$. Consequently, node $v$ becomes active at time $\tau$ with probability

$$1 - \prod_{u \in N(v) \cap S_{\tau-1}} (1 - p_{uv}),$$

in which case it is added to $S_\tau$. Each node may attempt to influence its neighbors at most once. The diffusion process terminates when $S_\tau = \emptyset$ for some $\tau$, and therefore completes in at most $n$ time steps. Let $(S_0, S_1, \ldots, S_{n-1})$ denote the resulting sequence of active node sets.

Given a seed set $S_0$, the *influence spread* is defined as $f(S_0) = \mathbb{E}[|\bigcup_{\tau=0}^{n-1} S_\tau|]$, that is, the expected number of active nodes at the end of the diffusion process. When it is useful to make the dependence on the edge probabilities explicit, we write $f(S_0, \mathbf{p})$. The function $f : 2^V \to \mathbb{R}_+$ is referred to as the *influence spread function*. The (offline) influence maximization problem takes as input the graph $G$ and a budget $k \in \mathbb{N}$, and seeks a seed set

$$S_0^{\text{opt}} \in \operatorname{argmax}_{S_0 \subseteq V : |S_0| \leq k} f(S_0, \mathbf{p}).$$

It is well known that under the IC model, $f(\cdot)$ is monotone and submodular, and that a $(1 - 1/e - \varepsilon)$ approximation can be achieved in polynomial time for any $\varepsilon > 0$ using a greedy algorithm (Kempe et al., 2003); this guarantee is tight unless P = NP (Feige, 1998).

**The Online Setting.** In the online influence maximization (OIM) problem studied in this paper, there is an underlying graph $G = (V, E)$ whose edge probability vector $\mathbf{p} \in (0, 1)^m$ is unknown. The learner interacts with the diffusion process over $T$ rounds, where $T \geq \max\{n, m\}$, and the influence probabilities remain fixed across all rounds. At each round $t \in [T]$, the learner selects a seed set $S_{t,0} \subseteq V$ with $|S_{t,0}| \leq k$. The independent cascade process is then realized according to the true probabilities $\mathbf{p}$.

After the diffusion terminates, the learner observes *node-level feedback*, namely the sequence of active node sets $(S_{t,0}, S_{t,1}, \ldots, S_{t,n-1})$. Using this feedback, the learner updates her knowledge of underlying diffusion process before making future decisions. Due to partial identifiability issues under node-level feedback, we allow the learner to estimate (and define) the diffusion process using certain aggregate set-level parameters $\mathbf{P}$ instead of estimating the edge-level probabilities $\mathbf{p}$ (more details in Section 2).

We will assume that the learner has access to an offline oracle or algorithm that can return an $\alpha$-approximate seed set given estimates $\widehat{\mathbf{P}}$.[2]

**Definition 1.1** ($\alpha$-Approximate Offline Oracle). An $\alpha$-approximation oracle takes as input $(G, k, \widehat{\mathbf{P}})$ where the parameters $\widehat{\mathbf{P}}$ define the diffusion process over graph $G$. This oracle returns a seed set $S_0 \subseteq V$ with $|S_0| \leq k$ s.t. $f(S_0, \widehat{\mathbf{P}}) \geq \alpha \cdot \max_{S_0' \subseteq V : |S_0'| \leq k} f(S_0', \widehat{\mathbf{P}})$. where $f(\cdot, \widehat{\mathbf{P}})$ is the influence spread function under the diffusion process defined by $\widehat{\mathbf{P}}$.[3]

Note that the above oracle is different than the pair oracle considered in prior work (Li et al., 2020; Zhang et al., 2022), which needs to perform joint optimization over seed sets and feasible edge probabilities.

Equipped with such an oracle, the performance of an OIM algorithm is measured in terms of the *expected cumulative $\alpha$-regret* over $T$ rounds, defined as

$$R(T) = \mathbb{E}\left[\alpha \cdot T \cdot f(S_0^{\text{opt}}, \mathbf{p}) - \sum_{t=1}^{T} f(S_{t,0}, \mathbf{p})\right].$$

Our goal is to design learning algorithms for OIM with sublinear regret in $T$.

## 1.2. Main Result and Techniques

We first make the following assumption that will be required in our main result.

**Assumption 1.2.** There exists a parameter $\gamma \in (0, 1)$ such that $\prod_{u \in N(v)} (1 - p_{uv}) \geq \gamma$ for all $v \in V$. In other words, the probability that $v$ remains **inactive** (despite its neighbors being active) is lower-bounded by $\gamma$.

**Theorem 1.3** (Informal). *Consider an instance of the OIM problem under the independent cascade model with node-level feedback. Suppose that the learner has access to an $\alpha$-approximate offline oracle. Then, there exists an online learning algorithm that achieves the following $\alpha$-regret after*

---

[2]If the oracle is randomized, one can also consider $(\alpha, \beta)$-approximation where $\alpha$ is the approximation factor and $\beta$ is the success probability; our results extend to this setting with little to no modifications.

[3]We overload the notation here by using $\widehat{\mathbf{P}}$ as an argument to $f$ instead of the edge probabilities $\mathbf{p}$. This is to emphasize that the diffusion process is defined by the parameters $\widehat{\mathbf{P}}$.

*T rounds:* $R(T) = \widetilde{O}\left(\gamma^{-1} n \sqrt{T} \sum_{v \in V} d_v^{3/2}\right)$ *where $n$ is the number of nodes, $d_v$ is the in-degree of node $v$, and $\widetilde{O}(\cdot)$ hides polylogarithmic factors in $n, T$ and $d_v$.*

To our knowledge, this is the first result that achieves sublinear regret for OIM under the independent cascade model with node-level feedback using only a standard offline oracle. In contrast, Zhang et al. (2022) obtain a regret bound of $\widetilde{O}(\gamma^{-1} n^{5/2} D \sqrt{T})$ by relying on a stronger *pair oracle*. Our result significantly improves the dependence on the number of nodes $n$, while avoiding the need for a specialized oracle. Moreover, the algorithm of Zhang et al. (2022) requires a burn-in phase of $O(\gamma^{-4} n D^3)$ rounds, which contributes an additional regret term. Our approach does not rely on any such burn-in period. Finally, it is known that under edge-level feedback, the worst-case regret for the IC model is $\widetilde{O}(n^3 \sqrt{T})$ (Wang & Chen, 2017). Since $\sum_v d_v = m \le n^2$ and $d_v \le D$, our bound matches the worst-case regret under edge-level feedback up to a factor of $\sqrt{D}/\gamma$, despite operating under the strictly weaker node-level feedback model.

*Remark* 1.4 (On Assumption 1.2). This assumption is standard in the OIM literature with node-level feedback (Zhang et al., 2022) and ensures strong convexity of the node-level pseudo-likelihood. Three points are worth noting. (i) $\gamma$ is a worst-case bound; a tighter analysis yields the per-node bound $\widetilde{O}(n \sqrt{T} \sum_v d_v^{3/2}/\gamma_v)$ with $\gamma_v := \prod_{u \in N(v)}(1 - p_{uv})$, and our pay-per-use stability lemma further downweights rare neighborhoods $S \subseteq N(v)$ by $Q_{S,v}(S_0)$. (ii) The dependence on $\gamma$ is only polynomial; empirically observed values (Table 2) are substantially larger than the worst-case bound. (iii) Some dependence on the link-function Lipschitz parameter is unavoidable in generalized linear bandits (Filippi et al., 2010); tighter instance-specific dependence is left to future work. We note that the adversarial bound of Pasteris et al. (2024) avoids $\gamma$ dependence entirely; however, our bound improves on theirs in graph-structural parameters whenever $\sum_v d_v^{3/2}/\gamma_v \lesssim \sqrt{nk}$, and the empirical comparison in Section 4 shows that the stochastic structure we exploit translates to substantially better practical performance.

**Our Techniques.** Our main technical contribution is to overcome the challenge of partial identifiability in estimation of edge-level probabilities, by focusing on estimating *set-level* aggregate parameters that are identifiable from limited node-level feedback. Specifically, for each node $v$ and a subset $S$ of its in-neighbors, we estimate an aggregate parameter that captures the combined influence of $S$ on $v$. However, there are now $2^d$ such parameters for a node with in-degree $d$, making it difficult to estimate all these parameters simultaneously.

Similar to the work of Zhang et al. (2022), we rely on a key observation that the activation probability of a node can be expressed as a generalized linear function of these aggregate parameters, and show that efficient estimation of these parameters is possible using (pseudo-)likelihood maximization. Although our estimation procedure is similar at a high level to that of Zhang et al. (2022), there are several crucial differences in how we use this estimation procedure within our overall algorithmic framework. First, unlike Zhang et al. (2022), who construct a confidence ball over edge-level parameters and rely on a pair oracle abstraction to jointly select optimistic seeds and parameters, we use this procedure to construct a separate confidence interval for each set-level activation probability. This confidence interval is used to construct optimistic upper confidence bounds (UCBs). Second, we perform constrained maximization to ensure that the estimated parameters correspond to valid set-level activation probabilities (i.e., lie in the range $[0, 1]$). Lastly, we introduce regularization into the estimation procedure, which eliminates the need for the costly burn-in phase required by prior work.

Given the *optimistic* set-level activation probabilities, we show that a standard offline oracle (Theorem 1.1) can be used to produce *optimistic seed sets* by exploiting the monotonicity of the influence maximization problem. Given these optimistic seeds we rely on the principle of optimism in the face of uncertainty to bound the regret in terms of the width of the confidence intervals. However, since the number of these set-level probabilities is exponential in the in-degree, we only send the necessary parameters to the oracle, and the optimistic set-level probabilities are computed on-the-fly within the oracle as needed.

The final technical challenge arises from the fact that, unlike in classical linear bandits where the learner directly controls which arms are pulled, our algorithm only selects seed sets, while the feedback–namely, which subsets of neighbors attempt to activate a node–is stochastic and determined by the diffusion process. This makes it impossible to directly enforce uniform exploration of all parameters. To address this issue, we adapt techniques from Agarwal et al. (2024) and establish a *stability lemma* that bounds regret in a *pay-per-use* manner: the regret contributed by a set-level parameter is proportional to the expected number of times it is *used* by the diffusion process.

## 2. Our Algorithm

In this section, we present our learning algorithm for OIM under the IC model with node-level feedback. The algorithm, termed NODE-GLB, is designed using the principle of *optimism in the face of uncertainty* and can be viewed as an upper-confidence-bound (UCB) method for a collection of generalized linear bandit (GLB) problems defined on *node neighborhoods*.

The following theorem (proved in Section 3) describes our

main result which shows that our algorithm achieves sublinear regret, assuming access to an $\alpha$-approximation oracle for offline influence maximization.

**Theorem 2.1.** *Consider the online influence maximization problem under the independent cascade model with node-level feedback. Suppose the learner has access to an $\alpha$-approximation oracle for the offline influence maximization problem, and let* $\mathtt{OPT} = \max_{|S| \leq k} f(S, \mathbf{p})$ *denote the optimal influence under the true edge probabilities* $\mathbf{p}$*. Then, with an appropriate choice of confidence parameters, the* NODE-GLB *algorithm satisfies the following $\alpha$-regret bound after $T$ rounds:*

$$R(T) \ = \ \widetilde{O}\Big(\tfrac{n\sqrt{T}}{\gamma} \sum_{v \in V} d_v^{3/2}\Big),$$

*where $n$ is the number of nodes, $\gamma$ is a lower bound on the probability of activation for any node, $d_v$ is the in-degree of node $v$, and $\widetilde{O}(\cdot)$ hides polylogarithmic factors in $n$ and $T$.*

In the remainder of this section, we provide an overview of our algorithm, and present its individual components in detail, including parameter estimation via least squares, construction of confidence bounds, and the simulation-based greedy oracle used for seed selection.

### 2.1. Algorithm Overview

As discussed in Section 1, the key challenge in online influence maximization with node-level feedback is that the learner does not observe which incoming edge causes a node to become active. As a result, individual edge-level influence probabilities cannot be estimated directly. To address this difficulty, our algorithm operates at the level of node neighborhoods and treats learning at each node as a GLB problem. For each node $v \in V$, we model its activation as a function of the subset of its in-neighbors that attempt to influence it at a given diffusion step. Under the IC model, if exactly the set $S \subseteq N(v)$ of neighbors attempts to activate $v$, then $P_{S,v} = 1 - \prod_{u \in S}(1 - p_{uv})$. Defining transformed parameters $\theta_{uv} = -\log(1 - p_{uv})$ yields the linear representation $-\log(1 - P_{S,v}) = \mathbf{1}_S^\top \boldsymbol{\theta}_v$, where $\boldsymbol{\theta}_v = (\theta_{uv})_{u \in N(v)}$ and $\mathbf{1}_S$ denotes the characteristic vector of set $S$. Crucially, we note that conditional on the active neighborhood $S$, the activation of node $v$ follows a Bernoulli distribution with mean $\mu(\mathbf{1}_S^\top \boldsymbol{\theta}_v) = 1 - \exp(-\mathbf{1}_S^\top \boldsymbol{\theta}_v)$, which induces a generalized linear model with the link function $\mu(z) = 1 - \exp(-z)$.

This forms the basis of our algorithmic approach: we maintain, for each node $v$, a separate generalized linear bandit instance to estimate the parameters $\boldsymbol{\theta}_v$ by observing activations of $v$ conditional on active neighborhoods $S$. We provide a high-level in Algorithm 1.

---

**Algorithm 1** NODE-GLB

1: **Input:** Graph $G = (V, E)$, seed budget $k$, rounds $T$
2: Initialize $\mathbf{M}_{v,0} \leftarrow \lambda \mathbf{I}$ for all $v \in V$, $\delta \leftarrow 1/(3nT)$
3: **for** $t = 1, \cdots, T$ **do**
4:     Calculate Gram matrices $\{\mathbf{M}_{v,t-1}\}_v$ per Eq. (2)
5:     $\{\widehat{\boldsymbol{\theta}}_{v,t}\}_{v \in V} = \text{ESTIMATE}(\mathbf{h}_{t-1})$
6:     $S_{t,0} \leftarrow \text{OFFLINEORACLE}(G, k, \{\boldsymbol{\theta}_v, \mathbf{M}_{v,t-1}\}_{v \in V})$
7:     Observe cascade $(S_{t,0}, S_{t,1}, \ldots)$ and set $\mathbf{h}_t$.
8: **end for**

---

### 2.2. Parameter Estimation from Node-Level Feedback

We begin by describing how to estimate the transformed parameters $\{\boldsymbol{\theta}_v\}_{v \in V}$ from the observed node-level cascade feedback (see Algorithm 2 for the pseudocode).

**Constructing Observations.** Fix a node $v \in V$. Each time during diffusion when $v$ is exposed to a set $S \subseteq N(v)$ of newly active in-neighbors during the diffusion process and has not yet activated, we record a single data point for node $v$. Formally, if there exists a round $r < t$ and a time step $\tau$ in the cascade such that $v$ is not active until time $\tau$ and $S = S_{r,\tau} \cap N(v)$, then we create a data point $(\boldsymbol{x}_{v,r,\tau}, y_{v,r,\tau})$ where: the feature vector is $\boldsymbol{x}_{v,r,\tau} = \mathbf{1}_S \in \{0,1\}^{d_v}$ and the response is the binary variable $y_{v,r,\tau} = \mathbb{I}[v$ becomes active at time step $\tau + 1]$. It is easy to observe that, under the IC model, conditional on $\boldsymbol{x}_{v,r,\tau}$, we have: $\mathbb{E}[y_{v,r,\tau} \mid \boldsymbol{x}_{v,r,\tau}] = \mu(\boldsymbol{x}_{v,r,\tau}^T \cdot \boldsymbol{\theta}_v)$.

**Estimation via Convex Quasi-Likelihood.** We estimate $\boldsymbol{\theta}_v$ using a convex quasi-likelihood as follows. We associate each data point $(\boldsymbol{x}_{v,r,\tau}, y_{v,r,\tau})$ with the per-sample loss

$$\ell_{v,r,\tau}(\boldsymbol{\theta}) = \exp(-\boldsymbol{x}_{v,r,\tau}^T \cdot \boldsymbol{\theta}) + (1 - y_{v,r,\tau}) \cdot \boldsymbol{x}_{v,r,\tau}^T \cdot \boldsymbol{\theta},$$

whose gradient is $(\mu(\boldsymbol{x}_{v,r,\tau}^T \cdot \boldsymbol{\theta}) - y_{v,r,\tau}) \cdot \boldsymbol{x}_{v,r,\tau}^T$. Given all data points collected for node $v$ up to time $t - 1$, we define $\widehat{\boldsymbol{\theta}}_{v,t}$ as the minimizer of the regularized empirical objective:

$$\widehat{\boldsymbol{\theta}}_{v,t} \in \operatorname{argmin}_{\boldsymbol{\theta} \in \Theta_v} \Big\{ \tfrac{\lambda}{2} \|\boldsymbol{\theta}\|_2^2 + \sum_{r < t} \sum_{\tau \in \tau_{v,r}} \ell_{v,r,\tau}(\boldsymbol{\theta}) \Big\} \quad (1)$$

where $\lambda > 0$ is a regularization parameter, $\tau_{v,r}$ denotes the set of time steps at which data points for node $v$ are generated in round $r$, and $\Theta_v = \{\boldsymbol{\theta} \in \mathbb{R}^{d_v} : \theta_u \geq 0, \forall u \in N(v), \sum_{u \in N(v)} \theta_u \leq \log(1/\gamma)\}$ is a compact and convex parameter set. Note that this objective is similar to the objective used in Zhang et al. (2022) except the regularization.

This formulation yields mean-zero noise at the true parameter and the estimation error can be controlled using standard self-normalized martingale concentration from Theorem B.4 (Abbasi-Yadkori et al., 2011). Given data from a generalized linear model, we define the corresponding Gram matrix:

$$\mathbf{M}_{v,t-1} \ = \ \lambda \mathbf{I}_{d_v} + \sum_{r < t} \sum_{\tau \in \tau_{v,r}} \boldsymbol{x}_{v,r,\tau} \boldsymbol{x}_{v,r,\tau}^T, \quad (2)$$

**Algorithm 2** ESTIMATE: Parameter Estimation from Node-Level Feedback

1: **Input:** Observation history $\mathbf{h}_{t-1} = \{(S_{r,0}, \ldots, S_{r,n-1})\}_{r=1}^{t-1}$
2: **Parameters:** Regularization $\lambda > 0$; parameter set $\boldsymbol{\Theta}_v \subseteq \mathbb{R}^{d_v}$ (compact, convex)
3: **for** each node $v \in V$ **do**
4:     Construct the set of data points $\mathcal{D}_{v,t-1} = \{(\boldsymbol{x}_{v,r,\tau}, y_{v,r,\tau})\}$ from $\mathbf{h}_{t-1}$ (see Section 2.2)
5:     Initialize Gram matrix $\mathbf{M}_{v,t-1} \leftarrow \lambda \mathbf{I}_{d_v}$
6:     **for** each data point $(\boldsymbol{x}, y) \in \mathcal{D}_{v,t-1}$ **do**
7:         $\mathbf{M}_{v,t-1} \leftarrow \mathbf{M}_{v,t-1} + \boldsymbol{x}\boldsymbol{x}^\top$
8:     **end for**
9:     Compute the estimate $\widehat{\boldsymbol{\theta}}_{v,t}$ per Eq. (1)
10: **end for**
11: **return** $\{\widehat{\boldsymbol{\theta}}_{v,t}, \mathbf{M}_{v,t-1}\}_{v \in V}$

This matrix captures the amount of information collected in different directions of the parameter space and will be used to construct confidence bounds.

### 2.3. Optimistic Set-Level Activation Probabilities

We now describe how to construct optimistic estimates of the set-level activation probabilities. These optimistic estimates form the basis of the exploration–exploitation tradeoff in our algorithm.

Fix a node $v \in V$ and a round $t$. For any subset $S \subseteq N(v)$, the quantity of interest is the linear form $\mathbf{1}_S^\top \boldsymbol{\theta}_v$, which determines the activation probability $P_{S,v} = 1 - \exp(-\mathbf{1}_S^\top \boldsymbol{\theta}_v)$. By standard arguments for linear bandits, one can construct an upper confidence bound on this linear form using the estimate $\widehat{\boldsymbol{\theta}}_{v,t}$ and the Gram matrix $\mathbf{M}_{v,t-1}$ as follows: $\mathbf{1}_S^\top \widehat{\boldsymbol{\theta}}_{v,t} + \|\mathbf{1}_S\|_{\mathbf{M}_{v,t-1}^{-1}} \cdot \beta_t(\delta)$. The exact form of $\beta_t(\delta)$ and the proof of this guarantee are deferred to Section 3 (see Lemma 3.1). Using the above upper confidence bound and the monotonicity of the mapping $x \mapsto 1 - \exp(-x)$, we define the optimistic estimate of $P_{S,v}$ at round $t$ as

$$\widehat{P}_{S,v}^{(t)} = 1 - \exp\left(-\mathbf{1}_S^\top \widehat{\boldsymbol{\theta}}_{v,t} - \|\mathbf{1}_S\|_{\mathbf{M}_{v,t-1}^{-1}} \cdot \beta_t(\delta)\right). \quad (3)$$

With high probability, this construction guarantees that $\widehat{P}_{S,v}^{(t)} \geq P_{S,v}$, $\forall S \subseteq N(v), \forall v \in V$. Thus, $\widehat{P}_{S,v}^{(t)}$ serves as a valid upper confidence bound on the true set-level activation probability. Indeed, we will supply $\{\widehat{\boldsymbol{\theta}}_v, \mathbf{M}_{v,t-1}\}_{v \in V}$ to the offline $\alpha$-approximation algorithm. *Remark* 2.2. The optimistic probabilities in (3) play a central role in the algorithm. They ensure that the offline algorithm is executed using stochastically dominating probabilities. This will turn out to be crucial in analyzing the "stability lemma" (see Lemma D.4). At the same time, as the Gram matrices $\mathbf{M}_{v,t-1}$ grow with additional observations, the confidence widths $\|\mathbf{1}_S\|_{\mathbf{M}_{v,t-1}^{-1}} \beta_t(\delta)$ shrink, causing the

optimistic estimates to converge to the true probabilities.

In Appendix C, we describe an oracle for the offline influence maximization problem that relies solely on $P_{S,v}$ values that are computed via the corresponding $\{\boldsymbol{\theta}_v\}_{v \in V}$ vectors. Specifically, we show give an $\alpha$-approximation oracle with $\alpha = 1 - 1/e$. On combining this result with Theorem 2.1, we complete the proof of Theorem 1.3.

## 3. Regret Analysis

In this section, we present the regret analysis of Algorithm 1, thereby proving Theorem 2.1. The proof follows a standard optimism-based template: we first establish a high-probability confidence bound for the quasi-likelihood estimator at each node, and then use this bound to control the gap between the influence achieved by the learning algorithm and that of the (approximate) offline benchmark. Our analysis crucially relies on the following *sampling lemma*. In a nutshell, the lemma quantifies the accuracy of the parameter estimates produced by Algorithm 2 in a self-normalized (ellipsoidal) norm, uniformly over all rounds and all feature vectors. See the self-contained Appendix D for all proofs.

**Lemma 3.1** (Sampling Lemma). *Fix a node $v \in V$, confidence $\delta \in (0, 1)$ and regularization parameter $\lambda > 0$. Let $\mathbf{M}_{v,t-1}$ be the design matrix for node $v$ at round $t$ defined in Equation (2). Under Assumption 1.2, with probability at least $1 - \delta$, for all rounds $t \geq 0$, we have simultaneously*

$$\forall \boldsymbol{x} \in \mathbb{R}^{d_v}, \quad |\boldsymbol{x}^\top(\widehat{\boldsymbol{\theta}}_{v,t} - \boldsymbol{\theta}_v^*)| \leq \beta_{v,t}(\delta) \cdot \|\boldsymbol{x}\|_{\mathbf{M}_{v,t-1}^{-1}}, \quad (4)$$

*where the confidence radius $\beta_{v,t}(\delta)$ is defined as:*

$$\beta_{v,t}(\delta) := \frac{1}{\gamma}\left(\sqrt{\lambda}\|\boldsymbol{\theta}_v^*\|_2 + \sqrt{2\log\left(\left(\frac{\lambda+td_v}{\lambda d_v}\right)^{d_v} \cdot \frac{1}{\delta}\right)}\right).$$

We analyze the regret of Algorithm 1 under a high-probability *good event*, denoted $\mathcal{G}$, under which all confidence bounds hold uniformly over nodes and rounds. Formally, let $\mathcal{G}$ denote the event that for all nodes $v \in V$, all rounds $t \in [T]$, and all feature vectors $\boldsymbol{x} \in \mathbb{R}^{d_v}$, $|\boldsymbol{x}^\top(\widehat{\boldsymbol{\theta}}_{v,t} - \boldsymbol{\theta}_v^*)| \leq \beta_{v,t}(\delta)\|\boldsymbol{x}\|_{\mathbf{M}_{v,t-1}^{-1}}$. By taking a union bound over all $v \in V$ and $t \in [T]$, we get the following.

**Lemma 3.2.** *With the choice $\delta = 1/(nT)^2$ in Lemma 3.1, the good event $\mathcal{G}$ occurs with probability at least $1 - 1/nT$.*

In the remainder of the analysis, we condition on the good event $\mathcal{G}$. A key consequence of the good event is that all set-level activation probabilities used by the algorithm are *optimistic*, as formalized in the following lemma.

**Lemma 3.3.** *Under the good event $\mathcal{G}$, for any round $t$ and any seed set $S \subseteq V$, the optimistic influence estimate dominates the true influence, and we have $\mathbf{P}^* \leq \widehat{\mathbf{P}}$.*

We first complete the proof assuming that $\mathcal{G}$ holds (this assumption is removed later). A central ingredient in the

analysis is the following *stability lemma*, which quantifies how perturbations in the local activation probabilities affect the resulting influence spread.

**Lemma 3.4** (Stability Lemma). *Let $\mathbf{P}^*$ denote the true probabilities induced by the parameter vectors $\{\boldsymbol{\theta}_v^*\}_{v \in V}$, and let $\widehat{\mathbf{P}}$ denote any alternative collection of set-level probabilities such that $\mathbf{P}^* \le \widehat{\mathbf{P}}$. Suppose that for every node $v \in V$ and every neighborhood $S \subseteq N(v)$, the activation probabilities satisfy $\big|\widehat{P}_{S,v} - P_{S,v}^*\big| \le \epsilon_{S,v}$. Let $S_0$ be the seed set returned by the $\alpha$-approximation oracle when run with probabilities $\widehat{\mathbf{P}}$, and let $S_0^{\mathrm{opt}} \in \arg\max_{|S| \le k} f(S, \mathbf{P}^*)$ be an optimal seed set under the true probabilities. Then,*

$$\alpha f(S_0^{\mathrm{opt}}, \mathbf{P}^*) - f(S_0, \mathbf{P}^*) \le n \sum_{v \in V} \sum_{S \subseteq N(v)} Q_{S,v}(S_0) \cdot \epsilon_{S,v}$$

*where $Q_{S,v}(S_0)$ denotes the probability that, under the diffusion process defined by $\mathbf{P}^*$ and seed set $S_0$, the set of active in-neighbors of node $v$ attempting to activate $v$ is exactly $S$.*

We now give a sketch of the proof of Theorem 2.1.

*Proof Sketch of Theorem 2.1.* Fix round $t$ and let $\mathbf{h}^{t-1}$ denote the history before selecting $S_{t,0}$. Under the good event $\mathcal{G}$ (Lemma 3.2), the per-node confidence bound of Lemma 3.1 implies that the optimistic set-level probabilities satisfy $\mathbf{P}^* \le \widehat{\mathbf{P}}^{(t)}$ coordinate-wise (Lemma 3.3). Let $S_{t,0}$ be the seed set returned by the $\alpha$-approximation oracle on $\widehat{\mathbf{P}}^{(t)}$, and define $r_t := \alpha f(S_0^{\mathrm{opt}}, \mathbf{P}^*) - f(S_{t,0}, \mathbf{P}^*)$.

**Step 1: Reduce regret to probability estimation error.** Since $g(x) = 1 - e^{-x}$ is 1-Lipschitz on $\mathbb{R}_+$, for any $v$ and $S \subseteq N(v)$, we have $\widehat{P}_{S,v}^{(t)} - P_{S,v}^* \le \min\big\{1, \ \big|\mathbf{1}_S^\top(\widehat{\boldsymbol{\theta}}_{v,t} - \boldsymbol{\theta}_v^*)\big| + \beta_{v,t}(\delta)\|\mathbf{1}_S\|_{\mathbf{M}_{v,t-1}^{-1}}\big\}$. Under the good event $\mathcal{G}$, we have have $|\mathbf{1}_S^\top(\widehat{\boldsymbol{\theta}}_{v,t} - \boldsymbol{\theta}_v^*)| \le \beta_{v,t}(\delta)\|\mathbf{1}_S\|_{\mathbf{M}_{v,t-1}^{-1}}$ and $\mathbf{P}^* \le \widehat{\mathbf{P}}$. Consequently, we apply the Lemma 3.4 with $\epsilon_{S,v}^{(t)} := \min\{1, 2\beta_{v,t}(\delta)\|\mathbf{1}_S\|_{\mathbf{M}_{v,t-1}^{-1}}\}$, which yields $\mathbb{E}[r_t \mid \mathbf{h}^{t-1}] \le n \sum_{v \in V} \sum_{S \subseteq N(v)} Q_{S,v}(S_{t,0}) \epsilon_{S,v}^{(t)}$. On summing over $t$ and using that $\beta_{v,t}(\delta)$ is non-decreasing:

$$R(T) \le 2n \sum_{v \in V} \beta_{v,T}(\delta) \cdot$$
$$\sum_{t, S \subseteq N(v)} \mathbb{E}\Big[Q_{S,v}(S_{t,0}) \min\Big\{1, \|\mathbf{1}_S\|_{\mathbf{M}_{v,t-1}^{-1}}\Big\}\Big]. \quad (5)$$

**Step 2: Convert $Q_{S,v}$ to realized observations.** Let $\mathcal{O}_{S,v}^{(t)} \in \{0,1\}$ indicate whether in round $t$ the diffusion under $\mathbf{P}^*$ produces an observation for node $v$ with feature vector $\mathbf{1}_S$. Then $Q_{S,v}(S_{t,0}) = \Pr(\mathcal{O}_{S,v}^{(t)} = 1 \mid \mathbf{h}^{t-1})$, and the tower property gives

$$\mathbb{E}\Big[Q_{S,v}(S_{t,0})\psi_{S,v}^{(t)}\Big] = \mathbb{E}\Big[\mathcal{O}_{S,v}^{(t)}\psi_{S,v}^{(t)}\Big]$$

for any $\mathbf{h}^{t-1}$-measurable $\psi_{S,v}^{(t)}$. Thus the inner sum in (5) is

$$\mathbb{E}\Big[\sum_{t=1}^T \sum_{S \subseteq N(v)} \mathcal{O}_{S,v}^{(t)} \min\Big\{1, \|\mathbf{1}_S\|_{\mathbf{M}_{v,t-1}^{-1}}\Big\}\Big].$$

**Step 3: Elliptical potential.** Along any realization path, $\mathbf{M}_{v,t} = \mathbf{M}_{v,t-1} + \sum_S \mathcal{O}_{S,v}^{(t)}\mathbf{1}_S\mathbf{1}_S^\top$ with $\mathbf{M}_{v,0} = \lambda I$ and $\|\mathbf{1}_S\|_2^2 \le d_v$. A Cauchy–Schwarz step together with the elliptical potential lemma (e.g., Abbasi-Yadkori et al. (2011, Lemma 11)) yields the pathwise bound $\sum_{t=1}^T \sum_{S \subseteq N(v)} \mathcal{O}_{S,v}^{(t)} \min\Big\{1, \|\mathbf{1}_S\|_{\mathbf{M}_{v,t-1}^{-1}}\Big\} \le \sqrt{Td_v} \cdot \sqrt{2d_v \log\Big(\frac{\lambda + Td_v}{\lambda d_v}\Big)}$. Substituting into (5) and plugging in $\delta = (nT)^{-2}$ gives the stated $\widetilde{O}(\cdot)$ bound.

**Removing conditioning.** Finally, $R(T) \le nT$ deterministically and $\Pr(\mathcal{G}^c) \le 1/(nT)$, so the contribution of $\mathcal{G}^c$ is at most 1, completing the proof. □

## 4. Experimental Results

We provide a summary of computational results to illustrate the empirical behavior of our algorithm NODE-GLB. The goal of these experiments is not to provide a large-scale empirical benchmark, but rather to validate our theoretical bounds. We provide a detailed and self-contained computational section in Appendix E.

**Baselines.** We compare NODE-GLB against three standard baselines: RANDOM, which selects $k$ nodes uniformly at random each round; HIGH-DEGREE, which selects the $k$ nodes with the largest in-degrees and is non-adaptive; and PUREEXPLOIT-NLF (PE-NLF), a greedy exploitation-only method that computes seed sets using edge-probability point estimates learned from node-level feedback, without explicit exploration. These baselines have been considered in prior work of Vaswani et al. (2015); see appendix for details. We additionally compare against MSE3 (Pasteris et al., 2024), the sum-max submodular bandit algorithm applied to online influence maximization via the live-edge reduction.

**Network datasets.** We evaluate our algorithms on two real-world and two synthetic networks whose characteristics are summarized in Table 1. We use the widely adopted Epinions and Flixter datasets. To stress test our learning algorithm and the baselines described earlier, we create synthetic instances based on Barabási–Albert (BA) graphs and Decoy Hub graphs, which are designed to decouple node degree from influence.

**Setting activation probabilities.** For the two real-world datasets and the Barabási–Albert graphs, true edge activation probabilities are assigned according to the *weighted cascade* model. Specifically, for each directed edge $e = (u, v)$,

*Table 1.* Network datasets used in experiments. All four networks have 1,000 nodes; for Epinions and Flixter, we use the subgraph induced by the top 1,000 highest-degree nodes.

| Dataset | Nodes | Edges | Avg. in-degree |
|---|---|---|---|
| Epinions | 1,000 | 73,422 | 73.42 |
| Flixter | 1,000 | 6,719 | 6.7 |
| BA | 1,000 | 4,975 | 4.98 |
| Decoy Hub | 1,000 | 36,800 | 36.8 |

we set $p_{uv} \propto \frac{1}{|N(v)|}$ and normalize so that the resulting probabilities lie in $(0, 0.2)$. This weighted cascade model is a standard choice for evaluating influence maximization algorithms when true probabilities are unknown and diffusion data are unavailable, and has been widely used in prior work (Kempe et al., 2003; Lei et al., 2015). For the *Decoy Hub Graphs*, probabilities are chosen to deliberately decouple node degree from influence strength (see Appendix E for the complete details).

**Implementation Details.** We compare our algorithm to the baseline on each of the networks described above. The number of seed nodes $k$ is set to $50$ for each experiment, and every experiment itself is repeated $5$ times. We conduct our computations using Python 3.13 with an Apple M3 processor and 16 GB 2133 MHz LPDDR3 memory.

We evaluate the regret of each algorithm against an offline (clairvoyant) seed set $S_0$ which is computed by supplying the true edge activation probabilities to the offline oracle. For updating the node-level parameters in our algorithm, we implement the estimator described in Algorithm 2. The pseudo-likelihood optimization is performed using the ADAM optimizer with an initial learning rate of $10^{-3}$. To improve stability over the long horizon $T$, we decay the learning rate as $10^{-3}/\sqrt{t}$, where $t$ denotes the round index. All parameter updates are projected onto the feasible set based on Assumption 1.2.

**Results.** We plot the average cumulative regret $R(t)/t$ incurred by the algorithms against time $t$. We set the time horizon to $T = 1000$ for real-world datasets and $T = 2000$ for synthetic instances. Results for real-world datasets and synthetic instances are shown in Figures 1 and 2.

We observe several consistent trends across all datasets. NODE-GLB achieves consistently low average cumulative regret across all experiments. After an initial exploration phase, its average regret decreases steadily over time, which is consistent with our theoretical regret bound. In contrast, the RANDOM baseline exhibits consistently high regret and does not improve with time, as expected. The HIGH-DEGREE heuristic performs competitively in cases where there is a positive correlation between degree and influence (also observed by Vaswani et al. (2015)); however,

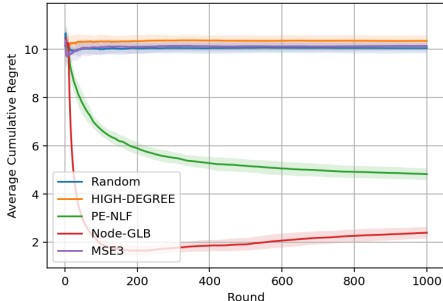

*(a)* Epinions Network

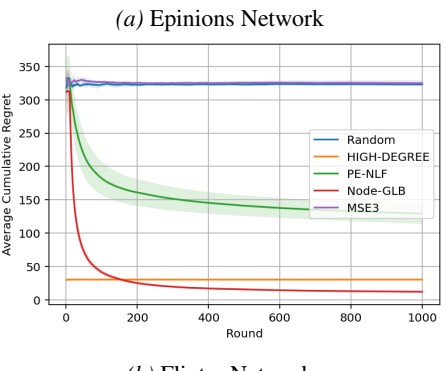

*(b)* Flixter Network

*Figure 1.* Average cumulative regret $R(t)/t$ on real-world datasets.

its regret curve remains essentially flat over time since it does not leverage any feedback. As expected, it performs poorly on instances based on the Decoy Hub Networks where there are high-degree decoys with limited influence. An interesting trend is observed in Figure 2b for the Decoy Hub Networks experiment, where our algorithm initially falls "victim" to decoy nodes due to large initial UCBs, but its regret improves significantly once the confidence intervals start to shrink. Finally, PE-NLF initially improves over non-adaptive baselines but consistently underperforms NODE-GLB. This gap highlights the importance of explicit optimism and uncertainty-aware exploration: although PE-NLF updates its estimates from feedback, its lack of principled exploration and "random credit assignment" causes it to prematurely commit to suboptimal seed sets.

Table 2 reports the empirical $\gamma$ values (substantially larger than the worst-case lower bound from edge-probability magnitudes alone) alongside average per-round wall-clock times. MSE3 behaves essentially like RANDOM at our horizons: its theoretically prescribed learning rate $\eta = \Theta(1/\sqrt{T})$ is on the order of $10^{-4}$ for our $(n, k, T)$, so the simplex distribution barely moves from uniform within $T$ rounds. Error bands across the 5 seeds are shown in all plots.

## 5. Conclusion

We address the problem of online influence maximization under the independent cascade model where influence prob-

| Network | Worst-case $\gamma$ | Observed $\gamma$ | Time/round (s) |
|---|---|---|---|
| Epinions | $1.1 \times 10^{-5}$ | $6.48 \times 10^{-3}$ | 2.52 |
| Flixter | $6.68 \times 10^{-2}$ | $1.72 \times 10^{-1}$ | 0.418 |
| BA | $1.88 \times 10^{-3}$ | $1.46 \times 10^{-1}$ | 0.197 |
| Decoy Hub | $1.98 \times 10^{-3}$ | $1.26 \times 10^{-1}$ | 0.190 |

*Table 2.* Worst-case and observed $\gamma$, and average wall-clock time per round for NODE-GLB.

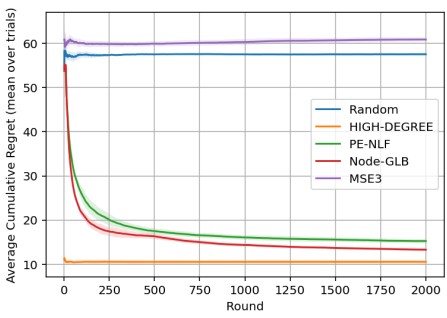

*(a)* Barabasi-Albert Graphs

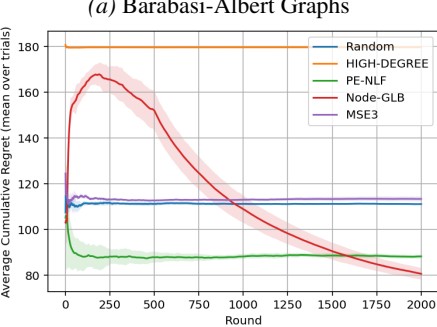

*(b)* Decoy Hub Graphs

*Figure 2.* Average cumulative regret $R(t)/t$ on synthetic instances.

abilities are unknown and feedback is limited to *node-level* activation outcomes (along with timestamps). Prior work on the problem has relied on pair oracles to achieve sublinear regret (Zhang et al., 2022), but these oracles are not known to admit efficient implementations. We design an algorithm that attains $\widetilde{O}(\sqrt{T})$ regret using only a standard offline oracle which is efficiently implementable. Our approach overcomes the challenge of partial identifiability of edge probabilities under limited feedback by estimating set-level activation probabilities through a reduction to generalized linear bandits. Our experimental results on synthetic and real-world networks validate our theoretical regret bound. An interesting direction for future work is to scale the experimental evaluation to networks with millions of nodes, study even weaker feedback models such as observing only the final activated set, and explore extensions to diffusion models beyond independent cascade.

# Acknowledgments

We thank the anonymous ICML reviewers for their careful feedback, which substantially improved the presentation of this work.

# Impact Statement

This paper presents work whose goal is to advance the field of Machine Learning. There are many potential societal consequences of our work, none which we feel must be specifically highlighted here.

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

## A. Further Related Work

**Influence Maximization.** The influence maximization problem was first formally studied by Kempe et al. (2003), who introduced the independent cascade and linear threshold diffusion models and showed that, under both models, the expected influence spread is a monotone submodular function of the seed set. This result implies that a simple greedy algorithm achieves a tight $(1 - 1/e)$-approximation, and that no polynomial-time algorithm can achieve a better approximation ratio unless $P = NP$ (Feige, 1998). Since then, a large body of work has focused on improving the scalability, robustness, and practical performance of offline influence maximization algorithms under various diffusion models and network assumptions; see, e.g., (Chen et al., 2010; Borgs et al., 2014; Tang et al., 2015).

**Online Influence Maximization with Edge-Level Feedback.** Early algorithms for learning influence probabilities assumed full-information *edge-level feedback*: after each diffusion round, the decision-maker observes the status of every edge involved in the cascade (i.e., which edges were live and which were not). Several subsequent works studied the online influence maximization problem under the weaker *semi-bandit edge-level feedback* model, in which the algorithm observes the outcomes of only those edges that are triggered by the chosen seed set (Chen et al., 2013; 2016; Wang & Chen, 2017; Wen et al., 2017; Wu et al., 2019).

Most of these works impose an additional *linear generalization* assumption: each edge $e \in E$ is associated with a known feature vector $x_e \in \mathbb{R}^d$, and the influence probability of $e$ is assumed to be (approximately) linear in the unknown parameter vector $\theta^* \in \mathbb{R}^d$. While this assumption enables generalization across edges, it leads to regret bounds that scale with the feature dimension $d$. For example, Wen et al. (2017) proposed the IMLINUCB algorithm with regret $\mathcal{O}(dn^3\sqrt{T})$, while the IMFB algorithm of Wu et al. (2019), which leverages network assortativity, achieves regret $\mathcal{O}(dn^{5/2}\sqrt{T})$.

A notable exception is the work of Chen et al. (2013; 2016), which does not rely on linear generalization. These papers model online influence maximization as a combinatorial multi-armed bandit with probabilistically triggered arms (CMAB-T) and apply the CUCB algorithm. However, the resulting regret bound contains a factor of $1/p^*$, where $p^*$ denotes the minimum probability that any edge is triggered under any feasible seed set. Since an edge can be triggered only if its source node becomes active, $p^*$ may be extremely small, leading to large regret guarantees. This issue was mitigated by Wang & Chen (2017), who introduced the triggering probability modulated (TPM) bounded smoothness condition and showed that online influence maximization satisfies this property, yielding regret bounds of order $\mathcal{O}(n^3\sqrt{T})$.

**Online Influence Maximization with Node-Level Feedback.** The node-level feedback model was introduced by Vaswani et al. (2015). Under this model, the learner observes only which nodes become active and at what times, but does not observe which neighbors caused each activation. Node-level feedback is often more realistic than edge-level feedback, as it requires substantially less information. Under node-level feedback, Li et al. (2020) proposed an algorithm achieving near-optimal $\mathcal{O}(\sqrt{T})$ regret under the linear threshold model, while Zhang et al. (2022) obtained a similar $\mathcal{O}(\sqrt{T})$ regret bound under the independent cascade model. However, both works rely on *pair oracles*, which jointly optimize over seed sets and admissible parameter estimates, rather than standard offline influence maximization oracles. While pair oracles are useful for theoretical analysis, they depart from the classical offline setting and can be difficult to realize in practice. Our work differs from these approaches by achieving sublinear regret under node-level feedback using only a standard offline influence maximization oracle. We note that an empirical comparison with Zhang et al. (2022) is not feasible: their algorithm relies on a pair oracle with no known efficient implementation, and the original version contained bugs with no corrected public implementation available.

A related line of work (Pasteris et al., 2024) studies *sum-max submodular bandits* in an adversarial setting. Conditioned on a live-edge graph realization, independent cascade influence is a coverage function and hence sum-max; treating each realized live-edge graph as the adversarially chosen matrix at round $t$, their framework does capture our problem, and their MSE3 algorithm provides an $\widetilde{O}(\sqrt{MKT})$ $(1 - 1/e)$-regret guarantee with $M = k$ and $K = n$. The resulting bounds, however, are incomparable to ours: their $\widetilde{O}(\sqrt{nkT})$ bound is governed by seed budget and node count, whereas ours captures the graph's degree structure via $\sum_v d_v^{3/2}$, exploits the stochastic IC structure for principled edge-probability learning rather than exponentiated-gradient updates on the simplex, and is compatible with a standard offline oracle. Empirically (see Section 4), MSE3's learning rate $\eta = \Theta(1/\sqrt{T})$ is small enough at our horizons that it behaves essentially like RANDOM.

**Network Inference.** An alternative approach to influence maximization with unknown parameters is to first learn the diffusion probabilities from historical cascade data and then apply offline influence maximization algorithms to the learned model. This learning task, known as *network inference*, has been extensively studied in the literature (Gomez-Rodriguez

et al., 2011; 2012; Du et al., 2012; Narasimhan et al., 2015; Chen et al., 2021). However, this two-stage approach does not account for the cost of learning and fails to balance exploration and exploitation when future diffusion cascades are influenced by the algorithm's own actions.

**Learning for Stochastic Optimization.** Finally, our work relates to a broader literature on learning for stochastic optimization, which seeks to understand the sample complexity and regret guarantees achievable when input distributions are unknown. Sample complexity bounds have been established for problems such as single-parameter revenue maximization (Cole & Roughgarden, 2014; Roughgarden & Schrijvers, 2016) and prophet inequalities (Correa et al., 2019). More recently, Guo et al. (2021) derived optimal sample complexity bounds for a broad class of stochastic optimization problems satisfying a strong monotonicity condition, which is slightly stronger than the monotonicity considered in our setting. Related learning-based approaches have also been studied in contextual stochastic optimization (Heuser & Kesselheim, 2025).

Some recent works also study stochastic optimization under semi-bandit or bandit feedback. In particular, Agarwal et al. (2024) propose a general framework for stochastic optimization with semi-bandit feedback and show that $\widetilde{O}(\sqrt{T})$ regret is achievable for a broad class of problems. Under their framework, one can also obtain $\widetilde{O}(\sqrt{T})$ regret guarantees for online influence maximization when *semi-bandit edge-level feedback* is available. More restrictive feedback models have also been studied. For example, Gatmiry et al. (2024) obtained online learning algorithms for Pandora's box and prophet inequality under pure bandit feedback, where the learner observes only the realized objective value of its policy. Their algorithms achieve regret bounds of order $O(n^3\sqrt{T}\log T)$ and $O(n^{4.5}\sqrt{T}\log T)$ for Pandora's box and prophet inequality, respectively.

## B. Preliminaries

We present some preliminaries that are required in our algorithm and proofs.

**Monotonicity.** We will use the following standard fact (Kempe et al., 2003) about the monotonicity of the influence maximization problem.

**Lemma B.1** (Monotonicity). *The (offline) influence maximization problem is monotone, i.e., for any graph $G = (V, E)$, seed set $S_0 \subseteq V$, and two edge probability vectors $\mathbf{p}$ and $\widehat{\mathbf{p}}$ such that $\mathbf{p} \le \widehat{\mathbf{p}}$ (coordinate-wise), we have*

$$f(S_0, \mathbf{p}) \le f(S_0, \widehat{\mathbf{p}}).$$

**Probability Distribution Metrics.** Let $\mathtt{TV\text{-}Dist}(P, \widehat{P})$ denote the *total variation distance* between discrete distributions $P$ and $\widehat{P}$. The total variation distance is half of the $\ell_1$ distance between the two distributions, i.e., $\mathtt{TV\text{-}Dist}(P, \widehat{P}) = \frac{1}{2} \cdot \|P - \widehat{P}\|_1$. The following standard result (see, for example, Lemma B.8 in Ghosal & Van der Vaart (2017)) bounds the total variation distance between product distributions.

**Lemma B.2.** *Given product distributions $\mathbf{P} = \{P_i\}_{i=1}^n$ and $\widehat{\mathbf{P}} = \{\widehat{P}_i\}_{i=1}^n$ over $n$ random variables, we have*

$$\mathtt{TV\text{-}Dist}(\mathbf{P}, \widehat{\mathbf{P}}) \le \sum_{i \in [n]} \mathtt{TV\text{-}Dist}(P_i, \widehat{P}_i).$$

Consider independent random variables $\mathtt{X}_1, \ldots, \mathtt{X}_n$ where $\mathsf{T}_i$ denotes the domain of $\mathtt{X}_i$. Let $h$ be a function from $\mathbf{T} = \mathsf{T}_1 \times \cdots \times \mathsf{T}_n$ to $[0, U]$; that is, $h$ is a function on the *outcomes* of the random variables that is bounded by $U$. Thus, for any $\mathbf{x} \in \mathbf{T}$, $h(\mathbf{x})$ denotes the value of $h$ on the outcome $\mathbf{x} = (x_1, \ldots x_n)$. Given a product distribution $\mathbf{Q}$ over the random variables, define $h(\mathbf{Q}) := \mathbb{E}_{\mathbf{x} \sim \mathbf{Q}}[h(\mathbf{x})]$. The following useful fact bounds the difference in function value at two different distributions.

**Lemma B.3.** *Given discrete distributions $\mathbf{P}$ and $\widehat{\mathbf{P}}$ over $n$ random variables $\mathtt{X}_1, \ldots, \mathtt{X}_n$, and a $[0, U]$ bounded function $h$ on the outcomes of these random variables (as above), we have*

$$|h(\mathbf{P}) - h(\widehat{\mathbf{P}})| \;\le\; U \cdot \mathtt{TV\text{-}Dist}(\mathbf{P}, \widehat{\mathbf{P}}).$$

**Self-Normalized Bounds.** For a positive semi-definite matrix $\mathbf{M} \in \mathbb{R}^{d \times d}$, we define the elliptical (Mahalanobis) norm of a vector $\boldsymbol{x} \in \mathbb{R}^d$ as $\|\boldsymbol{x}\|_{\mathbf{M}} := \sqrt{\boldsymbol{x}^\top \mathbf{M} \boldsymbol{x}}$. When $\mathbf{M}$ is invertible, we also use $\|\boldsymbol{x}\|_{\mathbf{M}^{-1}} := \sqrt{\boldsymbol{x}^\top \mathbf{M}^{-1} \boldsymbol{x}}$. Our proof for the

sampling lemma (Theorem D.1) uses the standard self-normalized bound for vector-valued martingales from Abbasi-Yadkori et al. (2011). We restate it here for convenience.

**Theorem B.4** (Theorem 1 from Abbasi-Yadkori et al. (2011)). *Let $\{\mathcal{F}_t\}_{t=0}^{\infty}$ be a filtration. Let $\{\eta_t\}_{t=1}^{\infty}$ be a real-valued stochastic process such that $\eta_t$ is $\mathcal{F}_t$-measurable and $\eta_t$ is conditionally $R$-sub-Gaussian for some $R \geq 0$, i.e., for all $\lambda \in \mathbb{R}$,*

$$\mathbb{E}[\exp(\lambda \eta_t) \mid \mathcal{F}_{t-1}] \leq \exp\left(\frac{\lambda^2 R^2}{2}\right).$$

*Let $\{\boldsymbol{x}_t\}_{t=1}^{\infty}$ be an $\mathbb{R}^d$-valued stochastic process such that $\boldsymbol{x}_t$ is $\mathcal{F}_{t-1}$-measurable. Assume that $\mathbf{V}$ is a $d \times d$ positive definite matrix. Further, for any $t \geq 0$, define*

$$\overline{\mathbf{V}}_t = \mathbf{V} + \sum_{s=1}^{t} \boldsymbol{x}_s \boldsymbol{x}_s^{\top} \qquad and \qquad \mathbf{S}_t = \sum_{s=1}^{t} \eta_s \boldsymbol{x}_s.$$

*Then, for any $\delta > 0$, with probability at least $1 - \delta$, for all $t \geq 0$,*

$$\|\mathbf{S}_t\|_{\overline{\mathbf{V}}_t^{-1}}^2 \leq 2R^2 \log\left(\frac{\det(\overline{\mathbf{V}}_t)^{1/2}}{\det(\mathbf{V})^{1/2}} \cdot \frac{1}{\delta}\right).$$

## C. Offline Influence Maximization with Set-Level Activation Probabilities

We now describe a standalone oracle for offline influence maximization under the independent cascade (IC) model when influence propagation is specified via set-level activation probabilities.

Recall that selecting the optimal seed set under the IC model is NP-hard. A classical result shows that the greedy algorithm, which iteratively adds the node with the largest marginal gain in expected influence, achieves a $(1 - 1/e)$-approximation when the influence spread function can be evaluated exactly. However, computing the expected influence of a seed set is itself intractable in general. A standard approach, introduced by Kempe et al. (2003), is to estimate influence spread via Monte Carlo simulation of the diffusion process. Most existing work assumes that diffusion is governed by known edge-level probabilities. In contrast, in our setting the diffusion process is specified by *set-level activation probabilities* $\{P_{S,v}\}$, where $P_{S,v}$ denotes the probability that node $v$ becomes active given that exactly the set $S \subseteq N(v)$ of its in-neighbors attempts to activate it.

**Simulating the IC Process with Set-Level Probabilities.** Given a seed set $S_0$, we simulate the IC diffusion process in discrete time steps. Let $S_\tau$ denote the set of active nodes after time step $\tau$, with $S_0$ initialized to the seed set. At each time step $\tau \geq 1$, for every node $v \notin S_{\tau-1}$ we compute the set $S = S_{\tau-1} \cap N(v)$ of active in-neighbors and activate $v$ independently with probability $P_{S,v}$. The diffusion process terminates when no new nodes become active. Repeating this simulation multiple times and averaging the resulting cascade sizes yields an unbiased estimator of the expected influence spread of $S_0$ under the specified set-level probabilities.

*Remark* C.1 (Implicit Representation of Activation Probabilities). The simulation-based oracle does not require explicit access to all set-level activation probabilities $\{P_{S,v}\}$ as input. Instead, it suffices to provide, for each node $v \in V$, a parameter vector $\boldsymbol{\theta}_v$ together with a positive definite matrix $\mathbf{M}_v$. Given these quantities, the oracle can compute $P_{S,v}$ (or $\widehat{P}_{S,v}$) on demand via the relation $P_{S,v} = 1 - \exp(-\mathbf{1}_S^{\top} \boldsymbol{\theta}_v)$ (or Equation (3)), without explicitly enumerating all subsets $S \subseteq N(v)$. This implicit representation is crucial for scalability, as the number of possible neighborhoods grows exponentially with the in-degree.

**Greedy Influence Maximization with Simulation.** Using the above simulation procedure as a subroutine, we implement a greedy influence maximization algorithm that iteratively constructs a seed set by selecting the node with the largest estimated marginal gain in influence. This procedure serves as a generic oracle for offline influence maximization given access to set-level activation probabilities. Algorithm 3 summarizes this simulation-based greedy oracle. Throughout the paper, we treat this oracle as a black box and assume it provides an $\alpha$-approximation to the optimal influence.

We conclude by stating the performance guarantee of the simulation-based greedy oracle. The guarantee combines the classical $(1 - 1/e)$-approximation of the greedy algorithm for submodular maximization with standard concentration bounds for Monte Carlo estimation of influence spread.

---

**Algorithm 3** OFFLINEORACLE: Greedy Influence Maximization

---

1: **Input:** Graph $G = (V, E)$, budget $k$, set-level probabilities $\{P_{S,v}\}$
2: Initialize seed set $S_0 \leftarrow \emptyset$
3: **while** $|\mathcal{S}| < k$ **do**
4: $\quad u \in \text{argmax}_{v \notin S_0} \{f(S_0 \cup \{v\}) - f(S_0)\}$
5: $\quad S_0 \leftarrow S_0 \cup \{u\}$
6: **end while**
7: **return** $S_0$

---

**Theorem C.2** (Offline Oracle Guarantee). *Fix a collection of set-level activation probabilities $\{P_{S,v}\}$ defining an independent cascade diffusion process. Let* $\text{OPT} = \max_{S:|S| \leq K} f(S)$ *denote the optimal expected influence spread under these probabilities. Then, Algorithm 3 returns a seed set $S_0$ satisfying $f(S_0) \geq (1 - 1/e) \cdot \text{OPT}$.*

Consequently, Algorithm 3 constitutes an $\alpha$-approximation oracle with $\alpha = 1 - 1/e$. Theorem C.2 establishes Algorithm 3 as a valid offline influence maximization oracle for arbitrary set-level activation probabilities. Combining Theorem 2.1 and Theorem C.2 completes the proof of Theorem 1.3. Throughout the remainder of the paper, we treat this oracle as a black box and focus on how it is invoked with different choices of $P_{S,v}$ by the online learning algorithm. The proof of Theorem C.2 follows standard arguments and is omitted.

## D. Regret Analysis

In this section, we present the regret analysis of Algorithm 1, thereby proving Theorem 2.1. The proof follows a standard optimism-based template: we first establish a high-probability confidence bound for the quasi-likelihood estimator at each node, and then use this bound to control the gap between the influence achieved by the learning algorithm and that of the (approximate) offline benchmark. Our analysis crucially relies on the following *sampling lemma*, which we prove first. In a nutshell, the lemma quantifies the accuracy of the parameter estimates produced by Algorithm 2 in a self-normalized (ellipsoidal) norm, uniformly over all rounds and all feature vectors.

**Lemma D.1** (Sampling Lemma). *Fix a node $v \in V$, confidence $\delta \in (0,1)$ and regularization parameter $\lambda > 0$. Let $\mathbf{M}_{v,t-1}$ be the design matrix for node $v$ at round $t$ defined in Equation (2), and let $\widehat{\boldsymbol{\theta}}_{v,t}$ be the corresponding estimate. Under Assumption 1.2, with probability at least $1 - \delta$, for all rounds $t \geq 0$, we have simultaneously*

$$\forall \boldsymbol{x} \in \mathbb{R}^{d_v}, \quad |\boldsymbol{x}^\top(\widehat{\boldsymbol{\theta}}_{v,t} - \boldsymbol{\theta}_v^*)| \leq \beta_{v,t}(\delta) \cdot \|\boldsymbol{x}\|_{\mathbf{M}_{v,t-1}^{-1}}, \tag{6}$$

*where the confidence radius $\beta_{v,t}(\delta)$ is defined as:*

$$\beta_{v,t}(\delta) := \frac{1}{\gamma}\left(\sqrt{\lambda}\|\boldsymbol{\theta}_v^*\|_2 + \sqrt{2\log\frac{1}{\delta} + 2d_v\log\left(\frac{\lambda + td_v}{\lambda d_v}\right)}\right). \tag{7}$$

*Here $\lambda > 0$ is the regularization parameter used in Algorithm 2.*

*Proof.* Recall that we associate each data point $(\boldsymbol{x}_{v,r,\tau}, y_{v,r,\tau})$ with the per-sample loss

$$\ell_{v,r,\tau}(\boldsymbol{\theta}) = \exp(-\boldsymbol{x}_{v,r,\tau}^T \cdot \boldsymbol{\theta}) + (1 - y_{v,r,\tau}) \cdot \boldsymbol{x}_{v,r,\tau}^T \cdot \boldsymbol{\theta},$$

and that $\boldsymbol{x}_{v,r,\tau}$ corresponds to the characteristic vector $\mathbf{1}_S$ for some set $S \subseteq N(v)$. Let $N_{S,v}^{(t)}$ denote the number of times, up to round $t$, that node $v$ is exposed to the neighborhood $S$ during the diffusion process, and let $\overline{N}_{S,v}^{(t)}$ denote the number of such times in which $v$ becomes active at the subsequent diffusion step. With this notation, the regularized pseudo log-likelihood objective for node $v$ at round $t$ can be written as:

$$\mathcal{L}_{v,t}(\boldsymbol{\theta}) = \frac{\lambda}{2}\|\boldsymbol{\theta}\|_2^2 + \sum_{S \subseteq N(v)} \left[N_{S,v}^{(t)} \exp\left(-\mathbf{1}_S^\top \boldsymbol{\theta}\right) + \left(N_{S,v}^{(t)} - \overline{N}_{S,v}^{(t)}\right)\mathbf{1}_S^\top \boldsymbol{\theta}\right].$$

Its gradient is

$$\nabla \mathcal{L}_{v,t}(\boldsymbol{\theta}) \;=\; \lambda\boldsymbol{\theta} + \sum_{S \subseteq N(v)} \left[ N_{S,v}^{(t)} \cdot \mu(\mathbf{1}_S^\top \boldsymbol{\theta}_v) - \overline{N}_{S,v}^{(t)} \right] \mathbf{1}_S, \tag{8}$$

where $\mu(x) = 1 - \exp(-x)$ is the link function.

Recall that $\widehat{\boldsymbol{\theta}}_{v,t}$ is defined as the minimizer of $\mathcal{L}_{v,t}(\boldsymbol{\theta})$ over a compact and convex parameter set $\Theta_v \subseteq \mathbb{R}^{d_v}$. Therefore, $\widehat{\boldsymbol{\theta}}_{v,t}$ satisfies the first-order optimality (KKT) condition

$$\left\langle \nabla \mathcal{L}_{v,t}(\widehat{\boldsymbol{\theta}}_{v,t}), \boldsymbol{\theta} - \widehat{\boldsymbol{\theta}}_{v,t} \right\rangle \;\geq\; 0, \qquad \forall \boldsymbol{\theta} \in \Theta_v. \tag{9}$$

Since the true parameter $\boldsymbol{\theta}_v^*$ lies in $\Theta_v$ by construction, the above condition holds in particular for $\boldsymbol{\theta} = \boldsymbol{\theta}_v^*$. Let $\Delta_{v,t} := \widehat{\boldsymbol{\theta}}_{v,t} - \boldsymbol{\theta}_v^*$. By (9) with $\boldsymbol{\theta} = \boldsymbol{\theta}_v^*$, we have

$$\left\langle \nabla \mathcal{L}_{v,t}(\widehat{\boldsymbol{\theta}}_{v,t}), \Delta_{v,t} \right\rangle \;\leq\; 0. \tag{10}$$

Next, decompose $\nabla \mathcal{L}_{v,t}(\widehat{\boldsymbol{\theta}}_{v,t}) = \nabla \mathcal{L}_{v,t}(\boldsymbol{\theta}_v^*) + \left( \nabla \mathcal{L}_{v,t}(\widehat{\boldsymbol{\theta}}_{v,t}) - \nabla \mathcal{L}_{v,t}(\boldsymbol{\theta}_v^*) \right)$. Taking inner products with $\Delta_{v,t}$ and using (10) yields

$$\left\langle \nabla \mathcal{L}_{v,t}(\widehat{\boldsymbol{\theta}}_{v,t}) - \nabla \mathcal{L}_{v,t}(\boldsymbol{\theta}_v^*), \Delta_{v,t} \right\rangle \;\leq\; - \left\langle \nabla \mathcal{L}_{v,t}(\boldsymbol{\theta}_v^*), \Delta_{v,t} \right\rangle. \tag{11}$$

Now, by the mean value theorem applied to the map $\boldsymbol{\theta} \mapsto \nabla \mathcal{L}_{v,t}(\boldsymbol{\theta})$, there exists $\tilde{\boldsymbol{\theta}}_{v,t}$ on the line segment joining $\boldsymbol{\theta}_v^*$ and $\widehat{\boldsymbol{\theta}}_{v,t}$ such that

$$\nabla \mathcal{L}_{v,t}(\widehat{\boldsymbol{\theta}}_{v,t}) - \nabla \mathcal{L}_{v,t}(\boldsymbol{\theta}_v^*) = \mathbf{H}_{v,t}(\tilde{\boldsymbol{\theta}}_{v,t}) \, \Delta_{v,t},$$

where the Hessian is

$$\mathbf{H}_{v,t}(\boldsymbol{\theta}) = \lambda \mathbf{I} + \sum_{S \subseteq N(v)} N_{S,v}^{(t)} \mu'(\mathbf{1}_S^\top \boldsymbol{\theta}) \, \mathbf{1}_S \mathbf{1}_S^\top.$$

Substituting into (11) gives

$$\Delta_{v,t}^\top \mathbf{H}_{v,t}(\tilde{\boldsymbol{\theta}}_{v,t}) \Delta_{v,t} \;\leq\; - \left\langle \nabla \mathcal{L}_{v,t}(\boldsymbol{\theta}_v^*), \Delta_{v,t} \right\rangle. \tag{12}$$

Under Assumption 1.2, for all $\boldsymbol{\theta} \in \Theta_v$ and all $S \subseteq N(v)$, we have $0 \leq \mathbf{1}_S^\top \boldsymbol{\theta} \leq \log(1/\gamma)$, hence $\mu'(\mathbf{1}_S^\top \boldsymbol{\theta}) = \exp(-\mathbf{1}_S^\top \boldsymbol{\theta}) \in [\gamma, 1]$. Therefore,

$$\mathbf{H}_{v,t}(\tilde{\boldsymbol{\theta}}_{v,t}) \;\succeq\; \lambda \mathbf{I} + \gamma \sum_{S \subseteq N(v)} N_{S,v}^{(t)} \mathbf{1}_S \mathbf{1}_S^\top \;=\; \gamma \, \mathbf{M}_{v,t-1} + (1-\gamma)\lambda \mathbf{I} \;\succeq\; \gamma \, \mathbf{M}_{v,t-1}. \tag{13}$$

where $\mathbf{M}_{v,t-1}$ is the design (Gram) matrix as defined in Equation (2). Combining (12) and (13) yields

$$\gamma \, \|\Delta_{v,t}\|_{\mathbf{M}_{v,t-1}}^2 \;\leq\; - \left\langle \nabla \mathcal{L}_{v,t}(\boldsymbol{\theta}_v^*), \Delta_{v,t} \right\rangle. \tag{14}$$

Using (8) and the identity $\mathbb{E}[\overline{N}_{S,v}^{(t)} \mid \mathcal{F}_{t-1}] = N_{S,v}^{(t)} \mu(\mathbf{1}_S^\top \boldsymbol{\theta}_v^*)$, we may write

$$\nabla \mathcal{L}_{v,t}(\boldsymbol{\theta}_v^*) = \lambda \boldsymbol{\theta}_v^* - \boldsymbol{\epsilon}_{v,t}, \qquad \boldsymbol{\epsilon}_{v,t} := \sum_{S \subseteq N(v)} \left( \overline{N}_{S,v}^{(t)} - N_{S,v}^{(t)} \mu(\mathbf{1}_S^\top \boldsymbol{\theta}_v^*) \right) \mathbf{1}_S,$$

where $\boldsymbol{\epsilon}_{v,t}$ is a martingale difference sum in $\mathbb{R}^{d_v}$. Substituting into (14) gives

$$\gamma \, \|\Delta_{v,t}\|_{\mathbf{M}_{v,t-1}}^2 \;\leq\; \langle \boldsymbol{\epsilon}_{v,t}, \Delta_{v,t} \rangle - \lambda \langle \boldsymbol{\theta}_v^*, \Delta_{v,t} \rangle. \tag{15}$$

Applying Cauchy–Schwarz and the inequality $|\langle a, b \rangle| \leq \|a\|_{\mathbf{M}^{-1}} \|b\|_{\mathbf{M}}$, we obtain $\langle \boldsymbol{\epsilon}_{v,t}, \Delta_{v,t} \rangle \leq \|\boldsymbol{\epsilon}_{v,t}\|_{\mathbf{M}_{v,t-1}^{-1}} \|\Delta_{v,t}\|_{\mathbf{M}_{v,t-1}}$, and $-\lambda \langle \boldsymbol{\theta}_v^*, \Delta_{v,t} \rangle \leq \lambda \|\boldsymbol{\theta}_v^*\|_2 \|\Delta_{v,t}\|_2 \leq \sqrt{\lambda} \, \|\boldsymbol{\theta}_v^*\|_2 \|\Delta_{v,t}\|_{\mathbf{M}_{v,t-1}}$, where the last step uses $\|\Delta\|_2 \leq \frac{1}{\sqrt{\lambda}} \|\Delta\|_{\mathbf{M}_{v,t-1}}$ since $\mathbf{M}_{v,t-1} \succeq \lambda \mathbf{I}$. Plugging into (15) and re-arranging yields

$$\|\Delta_{v,t}\|_{\mathbf{M}_{v,t-1}} \;\leq\; \frac{1}{\gamma} \left( \|\boldsymbol{\epsilon}_{v,t}\|_{\mathbf{M}_{v,t-1}^{-1}} + \sqrt{\lambda} \, \|\boldsymbol{\theta}_v^*\|_2 \right). \tag{16}$$

Consequently, for any $\boldsymbol{x} \in \mathbb{R}^{d_v}$,

$$|\boldsymbol{x}^\top(\widehat{\boldsymbol{\theta}}_{v,t} - \boldsymbol{\theta}_v^*)| \leq \|\boldsymbol{x}\|_{\mathbf{M}_{v,t}^{-1}} \|\Delta_{v,t}\|_{\mathbf{M}_{v,t-1}} \leq \frac{1}{\gamma} \|\boldsymbol{x}\|_{\mathbf{M}_{v,t-1}^{-1}} \left( \|\boldsymbol{\epsilon}_{v,t}\|_{\mathbf{M}_{v,t-1}^{-1}} + \sqrt{\lambda} \|\boldsymbol{\theta}_v^*\|_2 \right). \tag{17}$$

It remains to upper bound $\|\boldsymbol{\epsilon}_{v,t}\|_{\mathbf{M}_{v,t-1}^{-1}}$. Let $\mathcal{F}_{r,\tau-1}$ denote the $\sigma$-field generated by the entire history up to (and including) diffusion step $\tau - 1$ of round $r$. For each data point $(\boldsymbol{x}_{v,r,\tau}, y_{v,r,\tau})$ collected for node $v$, define the centered noise

$$\eta_{v,r,\tau} := y_{v,r,\tau} - \mu(\boldsymbol{x}_{v,r,\tau}^\top \boldsymbol{\theta}_v^*).$$

By construction, $\boldsymbol{x}_{v,r,\tau}$ is $\mathcal{F}_{r,\tau-1}$-measurable and $\mathbb{E}[\eta_{v,r,\tau} \mid \mathcal{F}_{r,\tau-1}] = 0$. Moreover, since $y_{v,r,\tau} \in \{0,1\}$ and $\mu(\cdot) \in [0,1]$, we have $|\eta_{v,r,\tau}| \leq 1$ almost surely, hence $\eta_{v,r,\tau}$ is conditionally 1-sub-Gaussian (i.e., Theorem B.4 holds with $R = 1$). Now enumerate the data points for node $v$ up to round $t$ in an arbitrary order as $\{(\boldsymbol{x}_s, y_s)\}_{s=1}^{m_{v,t}}$, with corresponding noises $\eta_s := y_s - \mu(\boldsymbol{x}_s^\top \boldsymbol{\theta}_v^*)$. Define

$$S_m := \sum_{s=1}^{m} \eta_s \mathbf{1}_s, \qquad \overline{V}_m := \lambda I + \sum_{s=1}^{m} \mathbf{1}_s \mathbf{1}_s^\top.$$

By definition of $\boldsymbol{\epsilon}_{v,t}$ and $\mathbf{M}_{v,t-1}$, we have $S_{m_{v,t}} = \boldsymbol{\epsilon}_{v,t}$ and $\overline{V}_{m_{v,t}} = \mathbf{M}_{v,t-1}$ (since repeated neighborhoods simply repeat the corresponding $\boldsymbol{x}_s = \mathbf{1}_S$). Therefore, applying Theorem B.4 with $V = \lambda I$, $d = d_v$, and $R = 1$ yields that with probability at least $1 - \delta$,

$$\|\boldsymbol{\epsilon}_{v,t}\|_{\mathbf{M}_{v,t-1}^{-1}}^2 \leq 2 \log \left( \frac{\det(\mathbf{M}_{v,t-1})^{1/2}}{\det(\lambda I)^{1/2}} \cdot \frac{1}{\delta} \right), \qquad \forall t \geq 0. \tag{18}$$

To simplify (18), note that $\mathbf{1}_s \in \{0,1\}^{d_v}$ with $\|\mathbf{1}_s\|_2^2 \leq d_v$, hence $\text{tr}\left( \sum_{s=1}^{m_{v,t}} \mathbf{1}_s \mathbf{1}_s^\top \right) = \sum_{s=1}^{m_{v,t}} \|\mathbf{1}_s\|_2^2 \leq m_{v,t} d_v$. Since for any PSD matrix $A$ of dimension $d_v$ we have $\det(\lambda I + A) \leq \left( \lambda + \frac{\text{tr}(A)}{d_v} \right)^{d_v}$, it follows that

$$\det(\mathbf{M}_{v,t-1}) = \det\left( \lambda I + \sum_{s=1}^{m_{v,t}} \mathbf{1}_s \mathbf{1}_s^\top \right) \leq \left( \lambda + \frac{m_{v,t} d_v}{d_v} \right)^{d_v} = (\lambda + m_{v,t})^{d_v}.$$

Using $\det(\lambda I) = \lambda^{d_v}$ and the crude bound $m_{v,t} \leq (t-1) d_v$ (since in any round $r$, node $v$ can generate at most $d_v$ data points), we obtain

$$\log \left( \frac{\det(\mathbf{M}_{v,t-1})^{1/2}}{\det(\lambda I)^{1/2}} \right) \leq \frac{d_v}{2} \log \left( \frac{\lambda + m_{v,t}}{\lambda} \right) \leq \frac{d_v}{2} \log \left( \frac{\lambda + t d_v}{\lambda} \right) = \frac{d_v}{2} \log \left( \frac{\lambda + t d_v}{\lambda d_v} \cdot d_v \right). \tag{19}$$

Plugging into (18) and absorbing constants yields the bound stated in Lemma D.1:

$$\|\boldsymbol{\epsilon}_{v,t}\|_{\mathbf{M}_{v,t-1}^{-1}} \leq \sqrt{2 \log \frac{1}{\delta} + 2 d_v \log \left( \frac{\lambda + t d_v}{\lambda d_v} \right)}. \tag{20}$$

Finally, combining (20) with (16) and then applying $|\boldsymbol{x}^\top(\widehat{\boldsymbol{\theta}}_{v,t} - \boldsymbol{\theta}_v^*)| \leq \|\boldsymbol{x}\|_{\mathbf{M}_{v,t-1}^{-1}} \|\widehat{\boldsymbol{\theta}}_{v,t} - \boldsymbol{\theta}_v^*\|_{\mathbf{M}_{v,t-1}}$ gives

$$|\boldsymbol{x}^\top(\widehat{\boldsymbol{\theta}}_{v,t} - \boldsymbol{\theta}_v^*)| \leq \frac{1}{\gamma} \left( \sqrt{\lambda} \|\boldsymbol{\theta}_v^*\|_2 + \sqrt{2 \log \frac{1}{\delta} + 2 d_v \log \left( \frac{\lambda + t d_v}{\lambda d_v} \right)} \right) \cdot \|\boldsymbol{x}\|_{\mathbf{M}_{v,t-1}^{-1}},$$

which is exactly the desired confidence bound. $\qquad \square$

We analyze the regret of Algorithm 1 under a high-probability *good event*, denoted $\mathcal{G}$, under which all confidence bounds hold uniformly over nodes and rounds. Formally, let $\mathcal{G}$ denote the event that for all nodes $v \in V$, all rounds $t \in [T]$, and all feature vectors $\boldsymbol{x} \in \mathbb{R}^{d_v}$,

$$\left| \boldsymbol{x}^\top(\widehat{\boldsymbol{\theta}}_{v,t} - \boldsymbol{\theta}_v^*) \right| \leq \beta_{v,t}(\delta) \|\boldsymbol{x}\|_{\mathbf{M}_{v,t-1}^{-1}}.$$

By taking a union bound over all $v \in V$ and $t \in [T]$, we get the following.

**Lemma D.2.** *With the choice $\delta = 1/(nT)^2$ in Lemma D.1, the good event $\mathcal{G}$ occurs with probability at least $1 - 1/nT$.*

In the remainder of the analysis, we condition on the good event $\mathcal{G}$. A key consequence of the good event is that all set-level activation probabilities used by the algorithm are *optimistic*, as formalized in the following lemma.

**Lemma D.3.** *Under the good event $\mathcal{G}$, for any round $t$ and any seed set $S \subseteq V$, the optimistic influence estimate dominates the true influence, and we have $\mathbf{P}^* \leq \widehat{\mathbf{P}}$ (coordinate-wise).*

*Proof.* Recall that the optimistic set-level activation probability is defined as

$$\widehat{P}_{S,v}^{(t)} = 1 - \exp\left(-\mathbf{1}_S^\top \widehat{\boldsymbol{\theta}}_{v,t} - \beta_{v,t}(\delta) \cdot \|\mathbf{1}_S\|_{\mathbf{M}_{v,t-1}^{-1}}\right).$$

Under the good event $\mathcal{G}$, we have

$$\mathbf{1}_S^\top \boldsymbol{\theta}_v^* \leq \mathbf{1}_S^\top \widehat{\boldsymbol{\theta}}_{v,t} + \beta_{v,t}(\delta) \cdot \|\mathbf{1}_S\|_{\mathbf{M}_{v,t-1}^{-1}} \qquad \forall v, S.$$

Let $h(x) = 1 - \exp(-x)$. Since $h(\cdot)$ is monotonically increasing, it follows that

$$\widehat{P}_{S,v}^{(t)} \geq 1 - \exp(-\mathbf{1}_S^\top \boldsymbol{\theta}_v^*) = P_{S,v}^*,$$

for all $v$ and $S \subseteq N(v)$, as desired. $\qquad\qquad\square$

We first complete the proof assuming that $\mathcal{G}$ holds (this assumption is removed later). A central ingredient in the analysis is the following *stability lemma*, which quantifies how perturbations in the local activation probabilities affect the resulting influence spread.

**Lemma D.4** (Stability Lemma). *Let $\mathbf{P}^*$ denote the true probabilities induced by the parameter vectors $\{\boldsymbol{\theta}_v^*\}_{v \in V}$, and let $\widehat{\mathbf{P}}$ denote any alternative collection of set-level probabilities such that $\mathbf{P}^* \leq \widehat{\mathbf{P}}$. Suppose that for every node $v \in V$ and every neighborhood $S \subseteq N(v)$, the activation probabilities satisfy $\left|\widehat{P}_{S,v} - P_{S,v}^*\right| \leq \epsilon_{S,v}$. Let $S_0$ be the seed set returned by the $\alpha$-approximation oracle when run with probabilities $\widehat{\mathbf{P}}$, and let $S_0^{\mathrm{opt}} \in \arg\max_{|S| \leq k} f(S, \mathbf{P}^*)$ be an optimal seed set under the true probabilities. Then,*

$$\alpha f(S_0^{\mathrm{opt}}, \mathbf{P}^*) - f(S_0, \mathbf{P}^*) \leq f(S_0, \widehat{\mathbf{P}}) - f(S_0, \mathbf{P}^*) \leq n \sum_{v \in V} \sum_{S \subseteq N(v)} Q_{S,v}(S_0) \cdot \epsilon_{S,v}$$

*where $Q_{S,v}(S_0)$ denotes the probability that, under the diffusion process defined by $\mathbf{P}^*$ and seed set $S_0$, the set of active in-neighbors of node $v$ attempting to activate $v$ is exactly $S$.*

We defer the proof of this lemma to Section D.1. We now complete the proof of Theorem 2.1 using Lemma D.4.

*Proof of Theorem 2.1.* Let $\mathbf{h}^{t-1}$ denote the history of observations up to the beginning of round $t$. At round $t$, the algorithm selects a seed set $S_{t,0}$ by invoking the offline oracle with the optimistic set-level probabilities $\widehat{\mathbf{P}}^{(t)}$. The instantaneous regret in round $t$ is then given by

$$r_t := \alpha f(S_0^{\mathrm{opt}}, \mathbf{P}^*) - f(S_{t,0}, \mathbf{P}^*),$$

where $S_0^{\mathrm{opt}}$ denotes an optimal seed set under the true probabilities. We begin by bounding the difference between optimistic and true activation probabilities. Recall that $g(x) = 1 - \exp(-x)$ is 1-Lipschitz on $\mathbb{R}_+$. For any node $v$ and neighborhood $S \subseteq N(v)$,

$$\widehat{P}_{S,v}^{(t)} - P_{S,v}^* = g\left(\mathbf{1}_S^\top \widehat{\boldsymbol{\theta}}_{v,t} + \beta_{v,t}(\delta)\|\mathbf{1}_S\|_{\mathbf{M}_{v,t-1}^{-1}}\right) - g(\mathbf{1}_S^\top \boldsymbol{\theta}_v^*) \leq \min\left\{1, \left|\mathbf{1}_S^\top(\widehat{\boldsymbol{\theta}}_{v,t} - \boldsymbol{\theta}_v^*)\right| + \beta_{v,t}(\delta)\|\mathbf{1}_S\|_{\mathbf{M}_{v,t-1}^{-1}}\right\}.$$

Under the good event $\mathcal{G}$ (Lemma D.2), we have $|\mathbf{1}_S^\top(\widehat{\boldsymbol{\theta}}_{v,t} - \boldsymbol{\theta}_v^*)| \leq \beta_{v,t}(\delta)\|\mathbf{1}_S\|_{\mathbf{M}_{v,t-1}^{-1}}$ and $\mathbf{P}^* \leq \widehat{\mathbf{P}}$ (Lemma D.3). Consequently, we apply the Lemma D.4 with $\epsilon_{S,v}^{(t)} := \min\{1, 2\beta_{v,t}(\delta)\|\mathbf{1}_S\|_{\mathbf{M}_{v,t-1}^{-1}}\}$, to obtain

$$\mathbb{E}_{\mathbf{h}^{t-1}}[r_t] \leq n \sum_{v \in V} \sum_{S \subseteq N(v)} \mathbb{E}_{\mathbf{h}^{t-1}}\left[Q_{S,v}(S_{t,0})\, \epsilon_{S,v}^{(t)}\right]$$

$$= n \sum_{v \in V} \sum_{S \subseteq N(v)} \mathbb{E}_{\mathbf{h}^{t-1}}\left[Q_{S,v}(S_{t,0}) \cdot \min\left\{1, 2\beta_{v,t}(\delta)\|\mathbf{1}_S\|_{\mathbf{M}_{v,t-1}^{-1}}\right\}\right].$$

We note that $r_t$, $S_{t,0}$, $\beta_{v,t}(\delta)$ and $\mathbf{M}_{v,t-1}$ all depend on the history $\mathbf{h}^{t-1}$; to keep notation simple, we drop this explicit dependence. Summing over $t = 1, \dots, T$ yields

$$R(T) = \sum_{t=1}^{T} \mathbb{E}_{\mathbf{h}^{t-1}}[r_t] \;\leq\; n \sum_{t=1}^{T} \sum_{v \in V} \sum_{S \subseteq N(v)} \mathbb{E}_{\mathbf{h}^{t-1}}\Big[ Q_{S,v}(S_{t,0}) \cdot \min\Big\{ 1,\, 2\beta_{v,t}(\delta)\|\mathbf{1}_S\|_{\mathbf{M}_{v,t-1}^{-1}} \Big\} \Big]. \tag{21}$$

Since $\beta_{v,t}(\delta)$ is non-decreasing in $t$, we may upper bound it by $\beta_{v,T}(\delta)$ to get

$$R(T) \;\leq\; 2n \sum_{v \in V} \beta_{v,T}(\delta) \cdot \sum_{t=1}^{T} \sum_{S \subseteq N(v)} \mathbb{E}_{\mathbf{h}^{t-1}}\Big[ Q_{S,v}(S_{t,0}) \cdot \min\Big\{ 1,\, \|\mathbf{1}_S\|_{\mathbf{M}_{v,t-1}^{-1}} \Big\} \Big]. \tag{22}$$

Thus, to complete the proof (under event $\mathcal{G}$), it suffices to control the quantity

$$\sum_{t=1}^{T} \sum_{S \subseteq N(v)} \mathbb{E}_{\mathbf{h}^{t-1}}\Big[ Q_{S,v}(S_{t,0}) \cdot \min\Big\{ 1,\, \|\mathbf{1}_S\|_{\mathbf{M}_{v,t-1}^{-1}} \Big\} \Big] \quad \text{for each } v \in V,$$

which we will do via a sample-path argument and the elliptical potential lemma.

**Summing over decision paths.** To bound the remaining term in (22), we switch from the conditional probabilities $Q_{S,v}(S_{t,0})$ to an equivalent sample-path representation (in the spirit of Agarwal et al. (2024)). Fix a node $v \in V$ and a round $t$. Given the random diffusion outcome in round $t$, define the indicator $\mathcal{O}_{S,v}^{(t)} \in \{0, 1\}$ to be 1 if, and only if, (during the cascade initiated from $S_{t,0}$ under the *true* diffusion model $\mathbf{P}^*$) there exists a diffusion step at which (i) $v$ is still inactive, and (ii) the set of newly active in-neighbors of $v$ is exactly $S$; in this case the algorithm records one observation for node $v$ with feature vector $\mathbf{1}_S$. By definition of $Q_{S,v}(S_{t,0})$, conditioning on the history $\mathbf{h}^{t-1}$,

$$Q_{S,v}(S_{t,0}) \;=\; \Pr\big( \mathcal{O}_{S,v}^{(t)} = 1 \,\big|\, \mathbf{h}^{t-1} \big), \tag{23}$$

and moreover $\mathcal{O}_{S,v}^{(t)}$ is measurable with respect to the randomness in the cascade at round $t$ (given $S_{t,0}$). Therefore, for any $\mathbf{h}^{t-1}$-measurable quantity $\psi_{S,v}^{(t)}$,

$$\mathbb{E}_{\mathbf{h}^{t-1}}\Big[ Q_{S,v}(S_{t,0})\, \psi_{S,v}^{(t)} \Big] = \mathbb{E}_{\mathbf{h}^{t}}\Big[ \mathcal{O}_{S,v}^{(t)}\, \psi_{S,v}^{(t)} \Big], \tag{24}$$

where $\mathbf{h}^t$ denotes the history extended to include the diffusion randomness in round $t$. Applying (24) with $\psi_{S,v}^{(t)} = \min\{1, \|\mathbf{1}_S\|_{\mathbf{M}_{v,t-1}^{-1}}\}$ and summing over $t$ and $S$ gives

$$\sum_{t=1}^{T} \sum_{S \subseteq N(v)} \mathbb{E}_{\mathbf{h}^{t-1}}\Big[ Q_{S,v}(S_{t,0}) \cdot \min\{1, \|\mathbf{1}_S\|_{\mathbf{M}_{v,t-1}^{-1}}\} \Big] = \mathbb{E}_{\mathbf{h}^T}\left[ \sum_{t=1}^{T} \sum_{S \subseteq N(v)} \mathcal{O}_{S,v}^{(t)} \cdot \min\{1, \|\mathbf{1}_S\|_{\mathbf{M}_{v,t-1}^{-1}}\} \right]$$

$$=: \mathbb{E}_{\mathbf{h}^T}\big[ Z_v(\mathbf{h}^T) \big]. \tag{25}$$

In words, $Z_v(\mathbf{h}^T)$ is the cumulative confidence width accrued along a *single realization path* of the algorithm and the cascades. Thus, bounding the regret reduces to bounding $Z_v(\mathbf{h}^T)$ pathwise, which we do next using the evolution of $\mathbf{M}_{v,t-1}$ and the elliptical potential lemma.

**Elliptical potential bound.** Fix a node $v \in V$ and a realization path $\mathbf{h}^T$. Recall that the design matrix for node $v$ evolves as

$$\mathbf{M}_{v,t} = \mathbf{M}_{v,t-1} + \sum_{S \subseteq N(v)} \mathcal{O}_{S,v}^{(t)} \mathbf{1}_S \mathbf{1}_S^\top, \qquad \mathbf{M}_{v,0} = \lambda \mathbf{I}_{d_v}.$$

Define the total number of observations collected for node $v$ along the path by

$$N_v(T) := \sum_{t=1}^{T} \sum_{S \subseteq N(v)} \mathcal{O}_{S,v}^{(t)}.$$

In each round, node $v$ can generate at most one observation per newly-active neighborhood, and there are at most $d_v$ in-neighbors; hence $N_v(T) \leq d_v T$ Since the confidence radii have already been factored out in (22), we bound the pathwise quantity

$$Z_v(\mathbf{h}^T) = \sum_{t=1}^{T} \sum_{S \subseteq N(v)} \mathcal{O}_{S,v}^{(t)} \cdot \min\left\{1, \|\mathbf{1}_S\|_{\mathbf{M}_{v,t-1}^{-1}}\right\}.$$

By Cauchy–Schwarz,

$$Z_v(\mathbf{h}^T) \leq \sqrt{N_v(T)} \cdot \sqrt{\sum_{t=1}^{T} \sum_{S \subseteq N(v)} \mathcal{O}_{S,v}^{(t)} \cdot \min\left\{1, \|\mathbf{1}_S\|_{\mathbf{M}_{v,t-1}^{-1}}^2\right\}}$$

$$\leq \sqrt{d_v T} \cdot \sqrt{\sum_{t=1}^{T} \sum_{S \subseteq N(v)} \mathcal{O}_{S,v}^{(t)} \cdot \min\left\{1, \|\mathbf{1}_S\|_{\mathbf{M}_{v,t-1}^{-1}}^2\right\}}, \tag{26}$$

where in the second inequality we used $N_v(T) \leq d_v T$. Next, we apply the standard elliptical potential lemma (Lemma 11 of Abbasi-Yadkori et al. (2011)). Since each observed feature vector is of the form $\mathbf{1}_S \in \{0,1\}^{d_v}$ and $\|\mathbf{1}_S\|_2^2 \leq d_v$, we obtain

$$\sum_{t=1}^{T} \sum_{S \subseteq N(v)} \mathcal{O}_{S,v}^{(t)} \cdot \min\left\{1, \|\mathbf{1}_S\|_{\mathbf{M}_{v,t-1}^{-1}}^2\right\} \leq 2\log\left(\frac{\det(\mathbf{M}_{v,T})}{\det(\mathbf{M}_{v,0})}\right) \leq 2 d_v \log\left(\frac{\lambda + T d_v}{\lambda d_v}\right). \tag{27}$$

The final inequality uses a similar reasoning as Eq. (19). Combining (26) and (27) yields the pathwise bound

$$Z_v(\mathbf{h}^T) \leq \sqrt{2 d_v T} \cdot \sqrt{d_v \log\left(\frac{\lambda + T d_v}{\lambda d_v}\right)}.$$

Taking expectations in (25) gives the same bound on $\mathbb{E}_{\mathbf{h}^T}[Z_v(\mathbf{h}^T)]$.

**Final bound.** Substituting the above pathwise bound on $Z_v(\mathbf{h}^T)$ into (22) yields, under the good event $\mathcal{G}$,

$$R(T) \leq 2n \sum_{v \in V} \beta_{v,T}(\delta) \cdot \mathbb{E}_{\mathbf{h}^T}[Z_v(\mathbf{h}^T)] \leq 2n \sum_{v \in V} \beta_{v,T}(\delta) \cdot \sqrt{2 d_v T} \cdot \sqrt{d_v \log\left(\frac{\lambda + T d_v}{\lambda d_v}\right)}.$$

We applied Lemma D.1 with $\delta = 1/(nT)^2$, and thus have

$$\beta_{v,T}(\delta) = \frac{1}{\gamma}\left(\sqrt{\lambda}\|\boldsymbol{\theta}_v^*\|_2 + \sqrt{4\log(nT) + 2 d_v \log\left(\frac{\lambda + T d_v}{\lambda d_v}\right)}\right).$$

Plugging this expression into the above expression and using $\sqrt{a+b} \leq \sqrt{a} + \sqrt{b}$ gives

$$R(T) \leq \frac{2n\sqrt{T}}{\gamma} \sum_{v \in V} d_v \sqrt{2\log\left(\frac{\lambda + T d_v}{\lambda d_v}\right)} \left(\sqrt{\lambda}\|\boldsymbol{\theta}_v^*\|_2 + \sqrt{4\log(nT)} + \sqrt{2 d_v \log\left(\frac{\lambda + T d_v}{\lambda d_v}\right)}\right)$$

$$\leq \frac{2n\sqrt{T}}{\gamma} \sum_{v \in V} \left[d_v \sqrt{\lambda}\|\boldsymbol{\theta}_v^*\|_2 \sqrt{2\log\left(\frac{\lambda + T d_v}{\lambda d_v}\right)} + 2 d_v \sqrt{2\log(nT)} \sqrt{\log\left(\frac{\lambda + T d_v}{\lambda d_v}\right)}\right.$$

$$\left. + 2 d_v^{3/2} \log\left(\frac{\lambda + T d_v}{\lambda d_v}\right)\right].$$

Using $\sum_v d_v = |E|$, we obtain the compact bound

$$R(T) \leq \tilde{O}\left(\frac{n\sqrt{T}}{\gamma}\left(\sqrt{\lambda}\sum_{v \in V} d_v \|\boldsymbol{\theta}_v^*\|_2 + |E| + \sum_{v \in V} d_v^{3/2}\right)\right) = \tilde{O}\left(\frac{n\sqrt{T}}{\gamma}\sum_{v \in V} d_v^{3/2}\right),$$

where the $\widetilde{O}(\cdot)$ hides polylogarithmic factors in $n$ and $T$.

So far, we assumed that event $\mathcal{G}$ holds. To remove the conditioning on $\mathcal{G}$, note that $R(T) \leq nT$ always and $\Pr(\mathcal{G}^c) \leq 1/(nT)$ by Lemma D.2, so the contribution from $\mathcal{G}^c$ is at most $nT \cdot \Pr(\mathcal{G}^c) \leq 1$. This completes the proof. $\qquad\square$

### D.1. Proof of the Stability Lemma (Lemma D.4)

We follow the proof strategy of the Stability Lemma of Agarwal et al. (2024) based on hybrid product distributions and a telescoping argument.

Recall that $\mathbf{P}^*$ denotes the true probabilities, and $\widehat{\mathbf{P}}$ denote an alternative collection of set-level probabilities such that $\mathbf{P}^* \leq \widehat{\mathbf{P}}$. Furthermore, $\left|\widehat{P}_{S,v} - P_{S,v}^*\right| \leq \epsilon_{S,v}$ for every node $v \in V$ and every neighborhood $S \subseteq N(v)$. Let $S_0$ be the seed set returned by the $\alpha$-approximation oracle when run with probabilities $\widehat{\mathbf{P}}$, and let $S_0^{\mathrm{opt}}$ be an optimal seed set under the true probabilities. First, note that since $\mathbf{P}^* \leq \widehat{\mathbf{P}}$, we have $f(S_0, \widehat{\mathbf{P}}) \geq f(S_0, \mathbf{P}^*)$ (see Lemma B.1). Moreover, $S_0$ is returned by the $\alpha$-approximation oracle under $\widehat{\mathbf{P}}$, so

$$f(S_0, \widehat{\mathbf{P}}) \;\geq\; \alpha \cdot \mathtt{OPT}(\widehat{\mathbf{P}}) \;\geq\; \alpha \cdot \mathtt{OPT}(\mathbf{P}^*) \;=\; \alpha f(S_0^{\mathrm{opt}}, \mathbf{P}^*).$$

Consequently,

$$\alpha f(S_0^{\mathrm{opt}}, \mathbf{P}^*) - f(S_0, \mathbf{P}^*) \;\leq\; f(S_0, \widehat{\mathbf{P}}) - f(S_0, \mathbf{P}^*).$$

Next, we bound the difference $|f(S_0, \widehat{\mathbf{P}}) - f(S_0, \mathbf{P}^*)|$. To do this, consider the diffusion process under the true model $\mathbf{P}^*$ and represent it as a (finite) rooted tree $\mathcal{T}$. Each node $u$ of $\mathcal{T}$ corresponds to a possible intermediate state of the cascade (i.e., the set of active nodes at a certain diffusion step). We index the nodes of $\mathcal{T}$ by a linear extension of the ancestor–descendant partial order (so ancestors have smaller indices).

For each internal node $u$, let $S(u)$ denote its active set. The random outcome at node $u$ is the next-step activation pattern produced by attempts from $S(u)$ to activate currently inactive nodes. Under $\mathbf{P}^*$ this outcome is distributed according to distribution $\{P_{N(v)\cap S(u),v}^*\}_v$, call this $D_u$; under $\widehat{\mathbf{P}}$ it is distributed according to $\{\widehat{P}_{N(v)\cap S(u),v}\}_v$, call this $E_u$. Importantly, $D_u$ and $E_u$ depend only on $S(u)$. Let $f_u$ denote the total number of active nodes at the leaf reached from $u$ (i.e., the eventual spread conditioned on reaching $u$); then $f_u \in [0, n]$. Thus, the influence spread can be written as an expectation of a bounded function over the product distribution of the per-node outcomes in $\mathcal{T}$.

Let the internal nodes of $\mathcal{T}$ be indexed as $u = 1, 2, \ldots, m$. Define hybrid product distributions by

$$\mathbf{H}^{(i)} := D_1 \times \cdots \times D_i \times E_{i+1} \times \cdots \times E_m, \qquad i = 0, 1, \ldots, m,$$

so that $\mathbf{H}^{(0)}$ corresponds to $\widehat{\mathbf{P}}$ and $\mathbf{H}^{(m)}$ corresponds to $\mathbf{P}^*$. By a telescoping sum,

$$f(S_0, \widehat{\mathbf{P}}) - f(S_0, \mathbf{P}^*) \;=\; f(S_0, \mathbf{H}^{(0)}) - f(S_0, \mathbf{H}^{(m)}) \;=\; \sum_{i=1}^{m} \Big( f(S_0, \mathbf{H}^{(i-1)}) - f(S_0, \mathbf{H}^{(i)}) \Big).$$

Fix an index $i$ and let $u$ denote the corresponding node of $\mathcal{T}$. Let $\mathcal{R}_u$ be the event that the diffusion process reaches node $u$. Since $\mathbf{H}^{(i-1)}$ and $\mathbf{H}^{(i)}$ differ only in the outcome distribution at node $u$, the expected spread is identical under the two hybrids conditioned on $\neg \mathcal{R}_u$. Conditioning on $\mathcal{R}_u$, the continuation of the process corresponds to a subtree rooted at $u$; the total value obtained in this subtree is bounded by $n$. Applying Lemma B.2 and Lemma B.3 yields

$$\left| f(S_0, \mathbf{H}^{(i-1)}) - f(S_0, \mathbf{H}^{(i)}) \right| \;\leq\; n \cdot \Pr_{\mathbf{P}^*}(\mathcal{R}_u) \cdot \mathtt{TV}(D_u, E_u).$$

At node $u$, each currently inactive vertex $v$ has at most one chance to activate from the newly active in-neighborhood $N(v) \cap S(u)$. The only difference between $D_u$ and $E_u$ is that for each such $v$, the Bernoulli activation probability is $P_{N(v)\cap S(u),v}^*$ under $D_u$ and $\widehat{P}_{N(v)\cap S(u),v}$ under $E_u$. Again, by Lemma B.2,

$$\mathtt{TV}(D_u, E_u) \;\leq\; \sum_{v \in V\,:\, N(v)\cap S(u) \neq \emptyset} \left| \widehat{P}_{N(v)\cap S(u),v} - P_{N(v)\cap S(u),v}^* \right|.$$

Combining the above two expressions and summing over $i = 1, \ldots, m$ gives

$$\left| f(S_0, \widehat{\mathbf{P}}) - f(S_0, \mathbf{P}^*) \right| \leq n \sum_{u \in \mathcal{T}} \Pr_{\mathbf{P}^*}(\mathcal{R}_u) \sum_{v:\, N(v) \cap S(u) \neq \emptyset} \left| \widehat{P}_{N(v) \cap S(u), v} - P^*_{N(v) \cap S(u), v} \right|.$$

Finally, we regroup terms by pairs $(v, S)$, where $S \subseteq N(v)$. For a fixed pair $(v, S)$, the coefficient of $\left| \widehat{P}_{S,v} - P^*_{S,v} \right|$ is precisely the probability (under $\mathbf{P}^*$) that the diffusion reaches a state in which the active set intersects $N(v)$ exactly on $S$ and then attempts to activate $v$ from that neighborhood. By definition, this probability is $Q_{S,v}(S_0)$. Using the assumption $\left| \widehat{P}_{S,v} - P^*_{S,v} \right| \leq \epsilon_{S,v}$, we conclude

$$\left| f(S_0, \widehat{\mathbf{P}}) - f(S_0, \mathbf{P}^*) \right| \leq n \sum_{v \in V} \sum_{S \subseteq N(v)} Q_{S,v}(S_0)\, \epsilon_{S,v}.$$

This completes the proof of the lemma.

## E. Experimental Results

We provide a summary of computational results to illustrate the empirical behavior of our algorithm NODE-GLB. The goal of these experiments is not to provide a large-scale empirical benchmark, but rather to validate our theoretical bounds.

**Baselines.**  We compare NODE-GLB against a representative set of baselines:

- RANDOM: selects a seed set of size $k$ uniformly at random at each round. This serves as a minimal baseline.

- HIGH-DEGREE: selects the $k$ nodes with the largest in-degrees in the graph at every round. This is a standard heuristic in influence maximization and does not adapt/learn over time.

- PUREEXPLOIT-NLF (PE-NLF): a greedy exploitation-only algorithm that maintains edge-level activation probability estimates using node-level feedback. At each round, the algorithm selects a seed set by solving the offline influence maximization problem using the current point estimates, without any explicit exploration. The edge-level parameters are updated as follows: whenever a node $v$ becomes active at diffusion step $\tau$, we identify the set of in-neighbors of $v$ that were newly activated at step $\tau - 1$ and could have caused the activation of $v$. One such in-neighbor is selected uniformly at random, and the corresponding edge activation count is incremented. This "random credit assignment" mechanism is a commonly used heuristic for learning from node-level feedback.

  MSE3.  We implement the algorithm of Pasteris et al. (2024) with their prescribed $\eta = \Theta(1/\sqrt{T})$. To stress-test sensitivity to tuning, we additionally run MSE3 with $\eta$ inflated by factors of $10^2$ and $10^4$ (equivalent to running with effective horizons $T/10^4$ and $T/10^8$ via the $1/\sqrt{T}$ scaling); even with this aggressive inflation, the simplex distribution moves slowly and the resulting regret remains close to RANDOM. In the plots, we report MSE3 under the best-performing $\eta$ across these settings.

We note that the first three baselines were also considered in prior work on online influence maximization with node-level feedback (Vaswani et al., 2015).

**Network datasets.**  We conduct experiments on two real-world and two synthetic networks. The real-world datasets were chosen to reflect common structures encountered in social networks, including heterogeneous degree distributions and moderate graph sizes. For each dataset, we report the number of nodes, edges, and average in-degree.

- **Epinions.** The Epinions trust network is a commonly used benchmark for influence maximization and diffusion-based learning problems. Nodes correspond to users of the Epinions consumer review platform, and a directed edge from user $u$ to user $v$ indicates that $v$ trusts $u$. The Epinions network consists of 75,879 nodes and 508,837 directed edges.

  To obtain a graph of manageable size, we compute the subgraph induced by the top 1,000 nodes with the highest degree. This subgraph has 73,422 directed edges and an average in-degree of 73.42.

- **Flixter.** The Flixster dataset represents a social network derived from user interactions on the Flixster movie rating platform. Nodes correspond to users, and edges capture social connections. This network consists of over 2.5 million nodes and over 9 million edges.

  Since the Flixster graph is originally undirected, we impose a direction on each edge to obtain a directed graph suitable for the Independent Cascade model. Specifically, for each undirected edge $\{u, v\}$, we direct the edge from the node with smaller degree to the node with larger degree (breaking ties arbitrarily). This orientation reflects the intuition that high-degree nodes are more likely to act as influential hubs, while lower-degree nodes are more likely to be influenced.

  As with Epinions, we compute the subgraph induced by the top $1\,000$ nodes with the highest degree. The resulting subgraph contains 6,719 edges and has an average in-degree of 6.7.

- **Barabasi-Albert Graphs.** We generate synthetic networks using the Barabási–Albert preferential attachment model. These graphs exhibit a power-law degree distribution and capture core structural properties of real-world social networks while allowing controlled scaling.

  Specifically, we generate graphs with 1,000 nodes and attachment parameter 5, resulting in $4,975$ total edges and an average in-degree of $4.98$. To obtain a directed influence network, we assign an orientation to each edge uniformly at random.

- **Decoy Hub Graphs.** We also design synthetic "hard" instances that are constructed to decouple node degree from influence strength, thereby highlighting the importance of learning activation probabilities from feedback.

  Concretely, the graph is composed of two disjoint classes of nodes: (i) a set of *decoy* nodes with very high out-degrees but uniformly small activation probabilities on their outgoing edges, and (ii) a set of *good* nodes with slightly smaller out-degrees but significantly larger activation probabilities. Edges from decoy nodes therefore create the appearance of high influence under degree-based heuristics, while contributing little to the actual diffusion process.

  We generate graphs with 1,000 nodes. The number of decoy nodes and good nodes is set to 300 and 100, respectively. Each decoy node has out-degree 100, while each good node has out-degree 50. The remaining nodes have an out-degree of 3; that is, the remainder of the graph is sparse. The resulting graph contains 36,800 directed edges and has an average in-degree of 36.8.

**Setting activation probabilities.** For the two real-world datasets and the Barabási–Albert graphs, true edge activation probabilities are assigned according to the *weighted cascade* model. Specifically, for each directed edge $e = (u, v)$, we set $p_{uv} \propto \frac{1}{|N(v)|}$ and normalize so that the resulting probabilities lie in $(0, 0.2)$. This weighted cascade model is a standard choice for evaluating influence maximization algorithms when true probabilities are unknown and diffusion data are unavailable, and has been widely used in prior work (Kempe et al., 2003; Lei et al., 2015).

For the *Decoy Hub Graphs*, probabilities are chosen to deliberately decouple node degree from influence strength. Edges leaving *good* nodes are assigned activation probabilities drawn uniformly from the interval $[0.1, 0.2]$, while edges leaving *decoy* nodes are assigned probabilities drawn uniformly from $[10^{-4}, 5 \times 10^{-4}]$. This sharp separation ensures that high out-degree alone is a poor proxy for influence, thereby stressing degree-based heuristics and highlighting the benefits of adaptive learning.

**Implementation Details.** We compare our algorithm to the baseline on each of the networks described above. The number of seed nodes $k$ is set to 50 for each experiment, and every experiment itself is repeated 5 times. We conduct our computations using Python 3.13 with an Apple M3 processor and 16 GB 2133 MHz LPDDR3 memory.

We use the widely adopted *Two-phase Influence Maximization* (TIM) algorithm by Tang et al. (2014) as an offline oracle (Theorem 1.1). In each call to TIM, we generate 5,000 reverse reachable sets to approximate the influence objective. Although our algorithm maintains confidence bounds over *set-level* activation probabilities, running TIM requires edge-level probabilities; we therefore convert set-level estimates to edge-level probabilities via the link function. We find that, empirically, this conversion does not degrade performance. The expected influence spread is estimated via 100 Monte Carlo simulations. The same oracle configuration is used across all algorithms to ensure a fair comparison. We evaluate the regret of each algorithm against an offline (clairvoyant) seed set $S_0$ which is computed by supplying the true edge activation probabilities to the offline oracle.

For updating the node-level parameters in our algorithm, we implement the estimator described in Algorithm 2. The pseudo-likelihood optimization is performed using the ADAM optimizer with an initial learning rate of $10^{-3}$. To improve

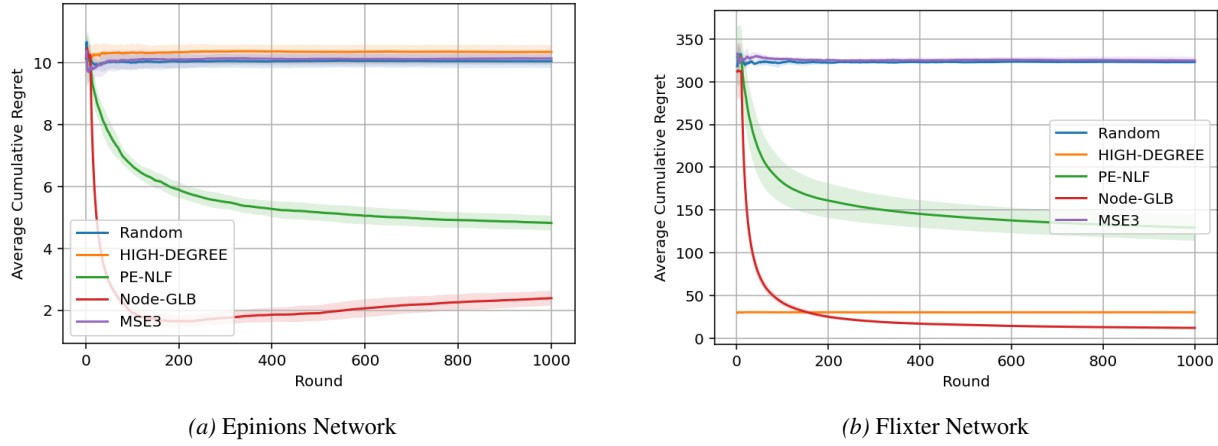

*(a)* Epinions Network                    *(b)* Flixter Network

*Figure 3.* Average cumulative regret $R(t)/t$ over time on real-world datasets.

stability over the long horizon $T$, we decay the learning rate as $10^{-3}/\sqrt{t}$. where $t$ denotes the round index. All parameter updates are projected onto the feasible set based on Assumption 1.2.

**Per-round cost and conversion.** Each round of NODE-GLB solves $n$ convex problems via ADAM (chosen over projected GD/L-BFGS purely for empirical speed in early rounds; all converged to the same solution) and one call to TIM. TIM dominates the cost and is shared across all algorithms. Since TIM uses edge-level probabilities, we convert set-level estimates via $p_{uv} = 1 - \exp(-\theta_{uv})$; this introduces mild estimation error which can only *worsen* empirical regret relative to the theoretical bound on set-level estimates, so reported results are a conservative evaluation.

**MSE3.** We implement Pasteris et al. (2024) with their prescribed $\eta = \Theta(1/\sqrt{T})$. To stress-test sensitivity to tuning, we additionally run MSE3 with $\eta$ inflated by factors of $10^2$ and $10^4$ (equivalent to running with effective horizons $T/10^4$ and $T/10^8$ via the $1/\sqrt{T}$ scaling); even with this aggressive inflation, the simplex distribution moves slowly and the resulting regret remains close to RANDOM.

**Results.** We plot the average cumulative regret $R(t)/t$ incurred by the algorithms against time $t$. We set the time horizon to $T = 1000$ for real-world datasets and $T = 2000$ for synthetic instances. Results for real-world datasets and synthetic instances are shown in Figures 3 and 4, respectively.

We observe several consistent trends across all datasets. NODE-GLB achieves consistently low average cumulative regret across all experiments. After an initial exploration phase, its average regret decreases steadily over time, which is consistent with our theoretical regret bound. In contrast, the RANDOM baseline exhibits consistently high regret and does not improve with time, as expected. The HIGH-DEGREE heuristic performs competitively in cases where there is a positive correlation between degree and influence (also observed by Vaswani et al. (2015)); however, its regret curve remains essentially flat over time since it does not leverage any feedback. As expected, it performs poorly on instances based on the Decoy Hub Networks where there are high-degree decoys with limited influence. An interesting trend is observed in Figure 4b for the Decoy Hub Networks experiment, where our algorithm initially falls "victim" to decoy nodes due to large initial UCBs, but its regret improves significantly once the confidence intervals start to shrink. Finally, PE-NLF initially improves over non-adaptive baselines but consistently underperforms NODE-GLB. This gap highlights the importance of explicit optimism and uncertainty-aware exploration: although PE-NLF updates its estimates from feedback, its lack of principled exploration and "random credit assignment" causes it to prematurely commit to suboptimal seed sets.

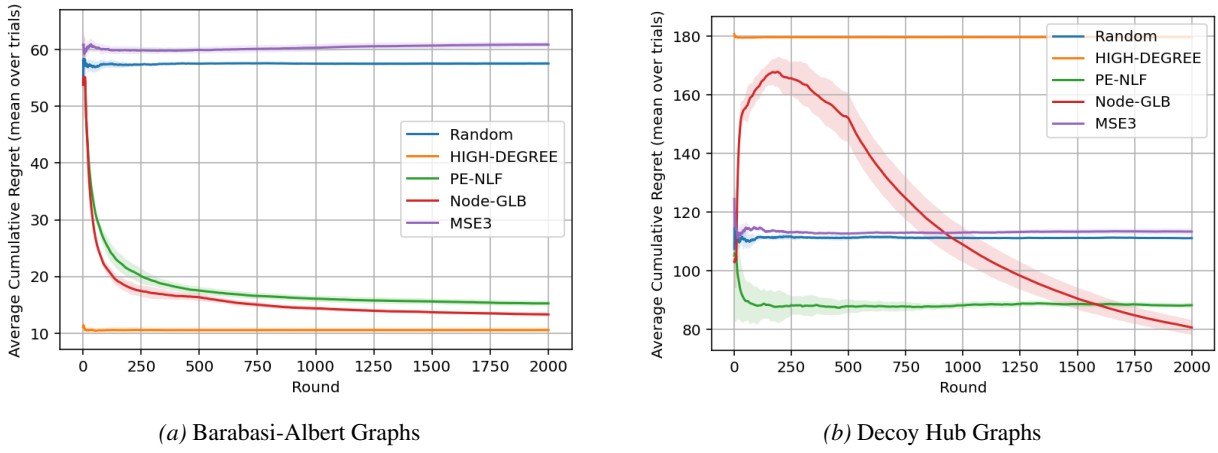

*(a)* Barabasi-Albert Graphs                         *(b)* Decoy Hub Graphs

*Figure 4.* Average cumulative regret $R(t)/t$ over time on synthetic instances.

