# OpenReview forum: "Efficient Online Influence Maximization under the Independent Cascade Model with Node-Level Feedback"
_ICML.cc/2026/Conference — ICML 2026 regular_

### Official Review · Reviewer_a6PW · 2026-03-10

**Soundness:** 3
**Presentation:** 3
**Significance:** 2
**Originality:** 3
**Overall Recommendation:** 5
**Confidence:** 4

**Summary:**

This paper solves online influence maximization under the independent cascade model with node-level feedback and unknown probabilities. It introduces NODE-GLB, the first algorithm to achieve regret using only a standard offline oracle. By estimating set-level activation probabilities via generalized linear bandits and employing optimistic confidence bounds, it eliminates the need for impractical pair oracles, improves theoretical bounds, and demonstrates efficacy in experiments.

**Compliance With Llm Reviewing Policy:**

Affirmed.

**Final Justification:**

I still feel that the results of (Zhang et al., 2021) have some impact on the findings of this submission.  However, it is possible that this impact is indeed not direct, so the results of this submission remain significant. I am willing to maintain my score.

**Key Questions For Authors:**

I find the results of the paper solid and would be willing to accept it if its main contribution is indeed independent. However, I remain concerned that the findings might be covered by the "Sum-max Submodular Bandits" paper. I would appreciate it if the authors could provide a clearer explanation of the relationship between the two works. If the contribution of this submission is partially covered by the prior work, I would consider lowering my score accordingly.

**Limitations:**

yes

**Strengths And Weaknesses:**

Strength：

It successfully resolves the open problem posed by prior work ( Zhang et al., 2021) of whether regret is achievable under node-level feedback using only a standard offline oracle. Their algorithm bypasses the need for the impractical pair oracle.

Weakness:

In my understanding, the result in the paper can be covered by the paper “Sum-max Submodular Bandits”(arXiv:2311.05975). In this paper, they discuss a more general submodular function class called sum-max function. Since coverage function is one of the sum-max function, and the influence function can be treated as the sum of coverage function (from the perspective of live-edge graph), influence function is also one of the sum-max function. They consider bandits setting which contains less information compare to the node-level feedback. I am not entirely certain of the relationship between the two, but it appears the authors have not taken note of this paper, so I fear the result of the submission has been covered in some sense.

---

> ### Author Rebuttal · Authors · 2026-03-30
>
> We thank the reviewer for the positive assessment and for raising this important question. We have carefully read the "Sum-max Submodular Bandits" paper (arXiv:2311.05975) and we believe our contributions are fully independent. We now explain why.
>
> **The influence spread function is not a sum-max function.**
> This is the key technical point. A sum-max function has the form $r(S) = \sum_{k \in [N]} \max_{i \in S} V_{k,i}$ for a *fixed* nonnegative matrix $V \in \mathbb{R}^{N \times K}$ (Definition 2.1 of Pasteris et al.). We acknowledge that a coverage function *is* sum-max: given sets $C_1, \ldots, C_K \subseteq [N]$, the coverage function $r(S) = |\bigcup_{i \in S} C_i|$ can be written as $\sum_{k \in [N]} \max_{i \in S} \mathbf{1}[k \in C_i]$, which is sum-max with the fixed binary matrix $V_{k,i} = \mathbf{1}[k \in C_i]$.
> The reviewer's intuition may therefore be that since each realization of the live-edge graph induces a coverage function, the IC influence function, which is an expectation of coverage functions, is also sum-max.
> However, this reasoning does not hold. While it is true that $f(S, p) = \mathbb{E}_G[|\text{reachable set from } S \text{ in } G|]$ where $G$ is a random live-edge graph, expressing this as a sum-max function would require a *fixed* matrix $V$ that does not depend on the random graph realization.
> No such fixed matrix exists: the reachability structure of $G$, and hence the matrix $V$ needed to represent the induced coverage function, varies across realizations.
> Taking the expectation over random graphs therefore moves the influence function *outside* the sum-max class, since the definition of Pasteris et al. requires $V$ to be fixed and deterministic.
> Concretely, this is also reflected in the computational complexity of the respective functions: sum-max functions can be evaluated in polynomial time by definition, whereas computing $f(S, p)$ exactly is #P-hard.
>
> **Differences in problem settings.**
> Pasteris et al. operate in a *nonstochastic/adversarial* setting where the reward function $r_t$ is chosen by an oblivious adversary at each round $t$. Our paper operates in a *stochastic* setting with a fixed but unknown edge probability vector $p$. These two settings require entirely different algorithmic and analytical machinery. In particular, their algorithm (MSE3) is based on exponentiated gradient ascent over a simplex of arm distributions, while ours is based on upper confidence bounds for generalized linear bandits combined with a stability lemma for the IC diffusion process.
>
> Furthermore, the feedback received is also different. In Pasteris et al., the learner observes a scalar reward $r_t(S_t)$ drawn from a known sum-max structure; in our setting, the feedback is not a scalar reward but an influence cascade revealing which nodes were activated and which of their neighbors they attempted to influence. This cascade feedback is strictly richer than scalar reward feedback and is essential for learning the unknown edge parameters. It is not at all clear whether one can achieve sublinear regret in our setting (without an $n^k$ dependence) by utilizing only the scalar rewards and not relying on the structure of the influence network.
>
> To summarize, given the fact that the influence function is not sum-max and there are key differences in settings, it is not possible to lift the machinery from Pasteris et al. to our setting and their results do not have any implications for our results.

---

> > ### Author Rebuttal · Reviewer_a6PW · 2026-04-02
> >
> > I still feel that the results of (Zhang et al., 2021) have some impact on the findings of this submission. According to the model in that paper, assuming the function at round t is $r_t$, it only requires that $r_t$ be submodular, and the player receives bandit feedback $r_t(S_t)$. For OIM, the feedback is the propagation process on the t-th live-edge graph, which corresponds to information about the coverage function defined by the live-edge graph, without needing to consider the influence function in general. However, it is possible that this impact is indeed not direct, so the results of this submission remain significant. I am willing to maintain my score.

---

### Official Review · Reviewer_b7MK · 2026-03-11

**Soundness:** 3
**Presentation:** 3
**Significance:** 2
**Originality:** 2
**Overall Recommendation:** 3
**Confidence:** 4

**Summary:**

This paper addresses the online influence maximization under the independent cascade model. The proposed method achieves sublinear regret under node-level feedback  without relying on specialized pair oracles. The paper provides better theoretical regret bound than existing method.

**Compliance With Llm Reviewing Policy:**

Affirmed.

**Key Questions For Authors:**

Please see the weaknesses.

**Limitations:**

Please clarify the limitation of the paper, espcially the application issues.

**Strengths And Weaknesses:**

Strengths

1. The paper addresses a well-motivated problem of online influence maximization with node-level feedback.

2. The idea of estimating set-level activation probabilities without explicit edge-level probabilities is clever and conceptually interesting.

3. The proofvis well-organized and technically sound.

4.  A stability lemma bounding how perturbations in local activation probabilities affect global influence spread.

5. Empirical validation on small synthetic networks.

Weaknesses

1. The main idea of the method is modifying the set probabilities in lines 231-247. In section 2.2, the framework is similar to Zhang et al. 2021. The method seems like a natural variant of Zhang's work, implying small technical contribution.

2. The assumption that all activation probabilities $P_{S,v} \ge \gamma$ for a global constant $\gamma > 0$ is extremely strong. In real large social networks, some users may have very large neighboring nodes, leading to quite small $\gamma$.
The regret bound scales as $1/\gamma$, resulting in large regret. In the experiment, how is the $\gamma$?

3. It seems that all real graphs have 1000 nodes in table 1, please confirm it.

4. The authors discuss much about Zhang's work, but do not compare with them in the experiment.

5. In the experiment, parameter influence and time cost are not investigate.

6. Theorem 1.3 is similar to Theorem 2.1.

7. Minor comments:
Reference of Zhang's work should be AAAI,2022 rather than arXiv.

---

> ### Author Rebuttal · Authors · 2026-03-30
>
> We thank the reviewer for their feedback and address each point in turn.
>
> **W1: Small technical contribution relative to Zhang et al. (2021).**
> We respectfully disagree. While our estimation procedure shares a high-level similarity with Zhang et al. (2021), the algorithmic and analytical contributions are substantially different. Zhang et al. construct a confidence *ellipsoid* over edge-level parameters and rely on a pair oracle to jointly optimize over seeds and parameters, an oracle with no known efficient implementation.
> By contrast, we construct separate UCBs for each set-level activation probability and show that a standard offline oracle suffices. This requires a fundamentally different regret analysis, including the pay-per-use stability lemma (Lemma 3.4) and the elliptical potential argument of Section 3, neither of which appear in Zhang et al. The result is a strictly weaker oracle requirement and no burn-in phase obtaining regret to the order of $\widetilde{O}(\sqrt{T})$.
>
> **W2: Assumption 1.2 and the value of $\gamma$.**
> We refer the reviewer to our detailed response to Reviewer qCnS (W3), where we discuss the practical values of $\gamma$ on our experimental networks and provide three mitigating arguments,
> including the pay-per-use accounting of Lemma 3.4 which ensures rare neighborhood configurations contribute negligibly to regret.
>
> **W3: All real graphs have 1,000 nodes.**
> Yes, this is correct and stated explicitly in Table 1 and Appendix E. For computational tractability we use the subgraph induced by the top 1,000 highest-degree nodes.
>
> **W4: No comparison with Zhang et al. (2021).**
> We refer the reviewer to our response to Reviewer qCnS (W2). In short, Zhang et al.'s algorithm requires a pair oracle with no known efficient implementation, contains bugs in its original version with no corrected public implementation available, making a fair empirical comparison infeasible.
>
> **W5: Parameter influence and time cost not investigated.**
> We refer the reviewer to our responses to Reviewer qCnS (W3) and (W4), where we discuss the influence of $\gamma$ and report wall-clock times per round respectively.
>
> **W6: Theorem 1.3 is similar to Theorem 2.1.**
> This is by design: Theorem 1.3 is the informal version stated in the introduction for accessibility, while Theorem 2.1 is the formal version with precise constants and conditions. This is a standard
> expository convention in theoretical machine learning papers.

---

> > ### Author Rebuttal · Reviewer_b7MK · 2026-04-04
> >
> > Thank you for your rebuttal.

---

### Official Review · Reviewer_qCnS · 2026-03-12

**Soundness:** 3
**Presentation:** 3
**Significance:** 3
**Originality:** 3
**Overall Recommendation:** 4
**Confidence:** 3

**Summary:**

This paper studies online influence maximization (OIM) under the independent cascade model where the learner observes only node-level feedback (which nodes activated and when, but not which edges caused activations). The main contribution is an algorithm called NODE-GLB that achieves $\\widetilde{O}(\\sqrt{T})$ cumulative $\\alpha$-regret using only a standard offline oracle for influence maximization. This resolves an open question, as prior work by Zhang et al. (2021) achieved the same regret rate but required a stronger pair oracle that jointly optimizes over seed sets and model parameters. The key technical idea is to estimate *set-level* activation probabilities via per-node generalized linear bandits, construct optimistic upper confidence bounds, and feed these to a simulation-based greedy oracle. A stability lemma bounds how perturbations in local activation probabilities affect global influence spread. Experiments on two real-world and two synthetic networks demonstrate that NODE-GLB achieves decreasing average regret over time, outperforming non-adaptive and exploitation-only baselines.

**Compliance With Llm Reviewing Policy:**

Affirmed.

**Final Justification:**

The paper makes a technically solid and reasonably original contribution to online influence maximization. The rebuttal and follow-up addressed several of my concerns. My main remaining concerns are the limited experimental scale and the lack of a direct comparison to the most relevant prior method, but overall I still view the work as a sound contribution with some practical limitations. I am therefore keeping my recommendation at Weak Accept.

**Key Questions For Authors:**

1. Can you provide a concrete characterization of $\\gamma$ for the experimental networks? For example, what is the empirical value of $\\min_v \\prod_{u \\in N(v)}(1-p_{uv})$ on the Epinions and Flixter subgraphs? How does the theoretical regret bound compare to the observed regret when instantiated with these values?

2. What is the per-round wall-clock time of NODE-GLB compared to the baselines? How does it scale with network size?

3. Could you implement Zhang et al. (2021) (with pair oracles) as an additional baseline to demonstrate the practical advantage of using standard oracles?

4. The paper uses ADAM for optimizing the convex quasi-likelihood objective (Eq. 1). Was a standard convex optimizer considered? Does the choice of optimizer affect the empirical performance or convergence?

5. For the conversion from set-level parameter estimates to edge-level probabilities when calling TIM: is this conversion exact under your parameterization, or does it introduce approximation error?

**Limitations:**

yes

**Strengths And Weaknesses:**

### Strengths

1. **Clean resolution of an open theoretical question**: The paper clearly identifies the gap in the literature (whether $\\widetilde{O}(\\sqrt{T})$ regret is achievable under node-level feedback with a standard offline oracle) and provides an affirmative answer.

2. **Elegant algorithmic design**: The reduction to per-node generalized linear bandits is natural and well-executed. The reparameterization using transformed parameters $\\theta_{uv} = -\\log(1 - p_{uv})$ turns node activation into a Bernoulli GLM, enabling the use of standard bandit tools (UCBs via self-normalized martingale bounds). The implicit representation of set-level probabilities (Remark C.1) is a nice computational insight.

3. **Novel stability lemma (Lemma 3.4/D.4)**: The hybrid argument over the diffusion tree that bounds the gap $|f(S_0, \\hat{\\mathbf{P}}) - f(S_0, \\mathbf{P}^*)|$ in terms of per-node activation probability errors is the key technical novelty.

4. **Clear writing and proof structure**: The paper is well-organized.

5. **Well-designed synthetic experiment**: The Decoy Hub construction is a clever stress test that decouples degree from influence, clearly demonstrating the advantage of learning-based approaches over degree heuristics.

### Weaknesses

1. **Limited experimental scale and evaluation**: All experiments use 1,000-node subgraphs, which is quite small for influence maximization. Real social networks have millions of nodes, and it is unclear how the algorithm scales. Moreover, the experimental plots lack error bars or confidence bands despite experiments being repeated 5 times.

2. **Missing comparison to the most relevant baseline**: Zhang et al. (2021) is the primary prior work this paper improves upon, yet no experimental comparison is provided. While the paper notes that Zhang et al.'s algorithm uses pair oracles (and had bugs in its original version), implementing their corrected algorithm for empirical comparison would provide valuable context.

3. **Assumption 1.2 may be restrictive**: The assumption that $\\prod_{u \\in N(v)}(1-p_{uv}) \\geq \\gamma > 0$ for all $v$ requires that every node has a non-trivial probability of remaining inactive. For nodes with large in-degree and moderate edge probabilities, $\\gamma$ can be exponentially small, making the regret bound $\\widetilde{O}(\\gamma^{-1} n \\sqrt{T} \\sum_v d_v^{3/2})$ vacuous. The paper would benefit from a more explicit discussion of when this assumption holds with reasonable $\\gamma$ values and what the effective regret bound looks like on the experimental networks.

4. **No discussion of computational cost**: Each round of NODE-GLB requires solving $n$ convex optimization problems (one per node) and calling the offline oracle (which itself runs Monte Carlo simulations). The per-round computational cost is not discussed or measured.

5. **Gap between theory and implementation**: The theory assumes exact minimization of the regularized quasi-likelihood (Eq. 1), but the implementation uses the ADAM optimizer with a decaying learning rate. ADAM is designed for non-convex optimization and provides no convergence guarantees for convex problems. Since the objective is convex, why not use a standard convex optimizer (e.g., projected gradient descent or L-BFGS)? Additionally, the theoretical oracle is the set-level greedy algorithm (Appendix C), but the implementation uses TIM, which requires edge-level probabilities. The conversion from set-level to edge-level estimates deserves discussion.

---

> ### Author Rebuttal · Authors · 2026-03-30
>
> We thank the reviewer for the thorough and constructive feedback. We address each point in turn.
>
> **W1: Limited experimental scale and missing error bars.**
> We agree that adding error bands would improve clarity and we will include them in the camera-ready version.
> Regarding scale: the goal of our experiments is not to provide a large-scale benchmark (as stated in Section 4) but rather to validate our theoretical regret bound.
> The consistent decrease of average cumulative regret $R(t)/t$ across all four networks is precisely the predicted behavior of an $\widetilde{O}(\sqrt{T})$ regret algorithm, and this trend is clearly visible even at $T = 1000$.
> We agree that scaling to larger networks is an interesting direction and we will leave this explicitly as a direction for future work in the camera-ready version.
>
> **W2: Missing comparison to Zhang et al. (2021).**
> We agree this comparison would be informative in principle. However, there are fundamental obstacles to a fair empirical comparison. First, the algorithm of Zhang et al. (2021) requires a *pair oracle*, an oracle that jointly optimizes over seed sets and feasible edge probability parameters, which is not known to admit an efficient implementation.
> This is precisely the limitation that motivates our work.
> Second, the original paper of Zhang et al. (2021) contained severe bugs that invalidated their main results (as acknowledged in their updated arXiv version), and no corrected public implementation of their algorithm is available.
> Efficiently implementing their corrected pair-oracle-based algorithm even for modest networks (with 1,000 nodes) is unclear.
> We will note these obstacles explicitly in the camera-ready version.
>
> **W3: Assumption 1.2 and the value of $\gamma$ on experimental networks.**
> We note first that Assumption 1.2 is standard in the OIM literature with node-level feedback and appears in prior work on this problem (Zhang et al., 2021). The assumption is needed to ensure strong convexity of the node-level pseudo-likelihood and to derive shrinking confidence regions.
>
> Regarding the practical value of $\gamma$ in our experimental setup: under the weighted cascade model, edge probabilities lie in $(0, 0.2)$, giving $\gamma \geq 0.8^{d_v}$ per node. For Flixter and Barabasi--Albert networks (average in-degree $6.7$ and $4.98$) this yields $\gamma \geq 0.27$ and $\gamma \geq 0.33$ respectively, which are reasonable values.
> For the denser Epinions network the worst-case bound is smaller, but we note three mitigating factors. First, $\gamma$ is a worst-case lower bound; most nodes have much smaller in-degree. Second, the regret scales only as $\gamma^{-1}$, a polynomial dependence. Third, the worst case requires *all* in-neighbors of a node to be simultaneously active, which is a rare event under typical cascade dynamics. Our pay-per-use stability lemma (Lemma 3.4) already accounts for this: rare neighborhood configurations $S \subseteq N(v)$ contribute negligibly to regret regardless of $\gamma$, since they are weighted by their realization probability $Q_{S,v}(S_0)$.
>
> We have re-run our experiments and report the empirically computed values of
> $$\gamma = \min_{v \in V} \prod_{u \in N(v)}(1-p_{uv})$$
> for each network. Please refer to our response to Reviewer qCnS for the *Table on worst case values of $\gamma$, observed values of $\\gamma$, and average wall-clock time per round for Node-GLB on each network.*
>
> **W4: Per-round computational cost.**
> We agree this discussion is missing and will add it to the camera-ready version. The per-round cost consists of
> (i) solving $n$ convex optimization problems via ADAM, and (ii) calling the offline oracle (TIM with
> 5,000 reverse reachable sets and 100 Monte Carlo simulations). Component (ii) dominates and is shared
> equally across all algorithms, ensuring a fair comparison. We will report average wall-clock times per round.
>
> **W5: TIM conversion.** Our theoretical guarantees are stated in terms of the set-level estimates
> $\hat P_{S,v}$. Converting to edge-level probabilities $p_{uv} = 1 - \exp(-\theta_{uv})$ for
> use in TIM introduces a mild estimation error, since our guarantees are on set-level parameters.
> However, this can only make the empirical regret *worse* relative to our theoretical bound.
> Our reported results therefore represent a conservative evaluation of Node-GLB's true performance, and the strong empirical results we observe are despite this introduced estimation error. We will include a couple of sentences about this in the camera-ready version.
>
> **Q4: Choice of ADAM optimizer.**
> We considered standard convex optimizers (projected gradient descent and L-BFGS) but found that
> ADAM converged to the same solution while being computationally faster in practice, particularly
> in earlier rounds when data is scarce. We will note this explicitly in the camera-ready version.

---

> > ### Author Rebuttal · Reviewer_qCnS · 2026-04-03
> >
> > Thank you for your response which partially resolved some of the issues. However, the experimental concerns (error bars, scale, runtime numbers) are all deferred to camera-ready without new evidence.

---

> > > ### Author Response · Authors · 2026-04-06
> > >
> > > We thank the reviewer for their response.
> > > We have  plotted the errors bars which can be seen using the links below.
> > >
> > > Epinions: https://anonymous.4open.science/r/OIM_ICML_2026-1D52/epinions.jpg
> > >
> > > Barabasi-Albert: https://anonymous.4open.science/r/OIM_ICML_2026-1D52/ba.jpg
> > >
> > > Flixter: https://anonymous.4open.science/r/OIM_ICML_2026-1D52/flixter.jpg
> > >
> > > Decoy Hub: https://anonymous.4open.science/r/OIM_ICML_2026-1D52/decoy.jpg
> > >
> > > We also included the following numbers for the $\gamma$ parameter and average wall-clock time per round in response to Reviewer XkJA.
> > >
> > >   | **Network** | **Worst-case $\gamma$** | **Observed $\gamma$** | **Time/Round(s)** |
> > >   |---|---|---|---|
> > >   | Epinions | 1.1 × 10⁻⁵ | 6.48 × 10⁻³ | 2.52 |
> > >   | Flixter | 6.68 × 10⁻² | 1.72 × 10⁻¹ | 0.418 |
> > >   | BA | 1.88 × 10⁻³ | 1.46 × 10⁻¹ | 0.197 |
> > >   | Decoy Hub | 1.98 × 10⁻³ | 1.26 × 10⁻¹ | 0.190 |
> > >
> > >   *Table: Worst case values of $\gamma$, Observed values of $\gamma$, and average wall-clock time per round for Node-GLB on each network.*
> > >
> > > We were able to perform a tighter analysis of our algorithm and achieve a bound of
> > > $\widetilde{O}(n\sqrt{T}\sum_{v\in V} \frac{d_v^{3/2}}{\gamma_v})$,
> > > where $\gamma_v$ is calculated per-node as $\gamma_v = \prod_{u \in N(v)}(1-p_{uv})$.
> > > Moreover, even $\gamma_v$ can be loose in practice because not all neighbors are active at the same time.
> > >
> > > We will add these to the final version of the paper.

---

### Official Review · Reviewer_XkJA · 2026-03-17

**Soundness:** 3
**Presentation:** 3
**Significance:** 2
**Originality:** 2
**Overall Recommendation:** 3
**Confidence:** 4

**Summary:**

The paper proposes a principled online influence maximization algorithm that achieves sublinear regret under node-level feedback while relying only on a standard, efficiently implementable offline oracle.

**Compliance With Llm Reviewing Policy:**

Affirmed.

**Final Justification:**

I increased my score from 2 (Reject) to 3 (Weak reject) following the authors' rebuttal.

**Key Questions For Authors:**

Refer to the strengths and weaknesses.

**Limitations:**

Refer to the strengths and weaknesses.

**Strengths And Weaknesses:**

1. A central pillar of the regret-proof is the **assumption that every node has a uniformly lower-bounded probability of remaining inactive even when all its neighbors attempt to activate it**. This condition is crucial for establishing strong convexity of the node-level pseudo-likelihood and for deriving shrinking confidence regions in the generalized linear bandit analysis. However, this assumption can be **quite strong in realistic diffusion settings**, where certain nodes—especially high-degree or highly influential ones—may activate with near certainty once exposed. If this lower bound is small or violated, the concentration guarantees used in the regret analysis would deteriorate significantly, calling into question whether sublinear regret could still be achieved.

2. Although the algorithm avoids explicitly enumerating the exponentially many set-level activation parameters by learning only those that arise during observed cascades, the regret analysis **implicitly assumes that the diffusion process will naturally expose all “important” neighborhoods frequently enough**. There is no explicit exploration mechanism that guarantees sufficient coverage of rare but potentially influential neighborhood configurations. Instead, the argument relies on a pay-per-use accounting of regret, where parameters that are not realized incur no cost. This logic depends heavily on benign stochastic properties of the diffusion process and could fail in networks where influential subsets appear rarely or only after long delays.

3. The stability lemma plays a critical role by translating local estimation errors in activation probabilities into global regret guarantees for influence spread. While the lemma is technically correct under the independent cascade model, it relies on **strong independence assumptions and a product-distribution structure over node activations**. The resulting bounds accumulate errors across all nodes and neighborhood configurations, scaling with the network size. This makes the analysis sensitive to even modest deviations from the assumed model, such as correlated activations, threshold effects, or richer behavioral dynamics, under which the argument would likely break down.

4. Even within the stated assumptions, **the sublinear regret guarantee conceals substantial dependence on structural graph parameters**, including the number of nodes and degree-based terms. In dense or heavy-tailed networks, these factors can dominate the bound, meaning the regret may remain large for any practical time horizon despite being asymptotically sublinear. As a result, the theoretical guarantee is best interpreted as an asymptotic possibility result rather than evidence of uniformly strong practical performance.

5. Finally, the analysis treats the offline influence maximization oracle as an **idealized component**, **assuming that optimistic set-level probabilities translate cleanly into seed sets that perform near-optimally under the true diffusion process**. In practice, such oracles rely on approximations, Monte Carlo estimation, and additional modeling assumptions, which introduce further error that is not reflected in the regret bound. While this gap does not undermine the internal logical correctness of the proof, it further limits the extent to which the theoretical sublinear regret guarantee can be viewed as robust or directly predictive of empirical behavior.

---

> ### Author Rebuttal · Authors · 2026-03-30
>
> We thank the reviewer for their comments.
>
> - **Regarding point 1 (the $\gamma$ assumption).**
>   This is a valid and important concern. We note first that Assumption 1.2 is standard in the OIM literature with node-level feedback and appears in prior work on this problem (Zhang et al., 2021). The assumption is needed to ensure strong convexity of the node-level pseudo-likelihood and to derive shrinking confidence regions.
>   Regarding the practical value of $\gamma$ in our experimental setup: under the weighted cascade model, edge probabilities lie in (0, 0.2), giving $\gamma \ge 0.8^{d_v}$ per node. For Flixter and Barábasi–Albert networks (average in-degree 6.7 and 4.98) this yields $\gamma \ge 0.27$ and $\gamma \ge 0.33$ respectively, which are reasonable values.
>   For the denser Epinions network the worst-case bound is smaller, but we note three mitigating factors. First, $\gamma$ is a worst-case lower bound; most nodes have much smaller in-degree. Second, the regret scales only as $\gamma^{-1}$, a polynomial dependence. Third, the worst case requires *all* in-neighbors of a node to be simultaneously active, which is a rare event under typical cascade dynamics. Our pay-per-use stability lemma (Lemma 3.4) already accounts for this: rare neighborhood configurations $S \subseteq N(v)$ contribute negligibly to regret regardless of $\gamma$, since they are weighted by their realization probability $Q_{S,v}(S_0)$.
>   We have re-run our experiments and report the empirically computed values of
>   $$\gamma = \min_{v \in V} \prod_{u \in N(v)}(1-p_{uv})$$
>   for each network.
>   We will include this in the camera-ready version.
>
>   | **Network** | **Worst-case $\gamma$** | **Observed $\gamma$** | **Time/Round(s)** |
>   |---|---|---|---|
>   | Epinions | 1.1 × 10⁻⁵ | 6.48 × 10⁻³ | 2.52 |
>   | Flixter | 6.68 × 10⁻² | 1.72 × 10⁻¹ | 0.418 |
>   | BA | 1.88 × 10⁻³ | 1.46 × 10⁻¹ | 0.197 |
>   | Decoy Hub | 1.98 × 10⁻³ | 1.26 × 10⁻¹ | 0.190 |
>
>   *Table: Worst case values of $\gamma$, Observed values of $\gamma$, and average wall-clock time per round for Node-GLB on each network.*
>
> - **Regarding point 2,** we respectfully note that this is not a weakness but rather the *central contribution* of our algorithm: the pay-per-use stability lemma (Lemma 3.4) precisely formalizes the observation that parameters which are rarely realized during diffusion contribute negligibly to regret, and the resulting bound is stated explicitly and in full detail in Theorem 2.1 with no hidden dependencies.
>   In regards to exploration, our algorithm automatically ensures that neighborhoods used by an optimal algorithm are sufficiently explored (else it would lead to high regret), while neighborhoods that are rare and not used by an optimal algorithm may not be explored (and in fact do not need to be explored).
>
> - **Regarding point 3,** the independence structure of the IC model is a standard and widely adopted assumption in virtually all bandit and influence maximization works; extending the analysis to correlated activations or threshold effects would require fundamentally different techniques and is well beyond the scope of this paper.
>
> - **Regarding point 4,** the regret upper bound is stated explicitly and in full detail in Theorem 2.1 with no hidden dependencies. The comment that "these factors can dominate the bound" is largely true for most of the (combinatorial) bandit literature where the regret bounds are not strong if the number of learnable parameters or arms is larger than the time-horizon.
>
> - **Regarding point 5,** the use of approximate oracles is a standard feature of all theoretical work in online learning and submodular optimization; for example, relying on an $\alpha$-approximate oracle leads to the standard notions of $\alpha$-regret bounds. The factor $\alpha$ can be a result of multiple different approximations including Monte Carlo estimation. We will make this point more explicit in the final version of the paper.

---

> > ### Author Rebuttal · Reviewer_XkJA · 2026-04-02
> >
> > I thank the authors for their rebuttal. It helps clarify the role of the inactivity parameter $\gamma$, and the distinction between worst‑case and empirically observed values is useful for understanding the experimental behavior. That said, my original concern remains only partially resolved. Even the reported observed values of $\gamma$ are quite small, and the regret bound scales polynomially in $1/\gamma$, which can result in very large constants despite being formally sublinear in $T$. Thus, while the asymptotic dependence on $T$ is sublinear, the guarantee still appears fragile and heavily model‑dependent, and may not be meaningful for realistic horizons in dense or heavy‑tailed networks.

---

> > > ### Author Response · Authors · 2026-04-06
> > >
> > > We thank the reviewer for their response.
> > > Recall that $\gamma$ is computed as, *worst-case over all nodes*, the probability that a node remains inactive when *all of its neighbors* attempt to activate it.
> > > Also recall the $\widetilde{O}(\gamma^{-1} n\sqrt{T}\sum_{v\in V} d_v^{3/2})$ bound reported in the paper.
> > >
> > > We were able to perform a tighter analysis of our algorithm and achieve a bound of
> > > $\widetilde{O}(n\sqrt{T}\sum_{v\in V} \frac{d_v^{3/2}}{\gamma_v})$,
> > > where $\gamma_v$ is calculated per-node as $\gamma_v = \prod_{u \in N(v)}(1-p_{uv})$.
> > > Moreover, as mentioned in our rebuttal, even $\gamma_v$ can be loose in practice because not all neighbors are active at the same time.
> > >
> > >
> > > While one can explore techniques to achieve an even better dependence on $\gamma$, the literature on generalized linear bandits (Filippi et al., 2010) suggests that some dependence on the Lipschitz parameter cannot be avoided whenever there is a link function applied to a linear function.
> > > Achieving an even tighter instance-specific dependence on $\gamma$ is a good direction for future work.

---

### Decision · Program_Chairs · 2026-04-30

**Decision:**

Accept (regular)

**Comment:**

The paper has clear theoretical contribution on online influence maximization (OIM), one of the representative instances in combinatorial online learning. The key contribution is to remove the dependency on pair-oracle in the prior work in OIM under IC model and node-level feedback. Instead, it only requires the standard optimization oracle. The technique to achieve this is via the estimation of UCB-like bounds on the influence to a node from a subset of edges, instead of UCB-like bounds on each edge individually. The reviewers acknowledge that this is a significant theoretical contribution to this research front. Since pair-oracle has also appeared in a number of other settings in combinatorial online learning, I also believe this contribution is significant and it may lead to further improvements in other problems that also depend on the use of pair-oracle.

Reviewers also raised a number of concerns. Authors addressed some of concerns but leave some others for the final version. One important concern is on the experiment validation and comparison with the existing one that uses the pair-oracle. I think it is important that the authors should give extensive empirical evaluation in the final version. Of course the authors explained that the pair-oracle is hard to implement. But this does not prevent the authors to propose some heuristic method to heuristically implement the pair-oracle in some sense, for comparison purpose. For example, one possible way to deal with simultaneous optimization of both the seed set and the parameters in the pair-oracle is to turn it into an iterative one, optimize one while fixing the other, and then vice versa. This at least give some heuristic way for a empirical comparison.

Overall, I believe that the authors made a significant theoretical contribution in OIM by replacing the hard-to-implement pair-oracle with a standard oracle.   It potentially also has impact in other combinatorial online learning settings. Therefore, I would like to recommend this paper for acceptance.